# Chondroitin 6-sulfate represses keratinocyte proliferation in mouse skin, which is associated with psoriasis

Kazuyuki Kitazawa [1], Satomi Nadanaka [1], Kenji Kadomatsu[2] & Hiroshi Kitagawa [1✉]

Chondroitin sulfates are implicated in epidermal biology, but functional significance of chondroitin sulfates remains unclear. Here, we report that chondroitin 6-sulfate is important for the maintenance of epidermal homeostasis. Mice deficient in chondroitin 6-O-sulfo-transferase-1 (C6st-1), which is involved in biosynthesis of chondroitin 6-sulfate, exhibited keratinocyte hyperproliferation and impaired skin permeability barrier function. Chondroitin 6-sulfate directly interacted with the EGF receptor and negatively controlled ligand-induced EGF receptor signaling. Normal function of hyperproliferative *C6st-1*-knockout mouse-derived keratinocytes was rescued by treatment with exogenous chondroitin 6-sulfate. Epidermal hyperplasia, induced using imiquimod, was more severe in *C6st-1*-knockout mice than in *C6st-1* wild-type mice. Taken together, these findings indicate that chondroitin 6-sulfate represses keratinocyte proliferation in normal skin, and that the expression level of *C6st-1* may be associated with susceptibility to psoriasis.

---

[1] Laboratory of Biochemistry, Kobe Pharmaceutical University, Higashinada-ku, Kobe 658-8558, Japan. [2] Department of Biochemistry, Nagoya University Graduate School of Medicine, 65 Tsurumai-cho, Showa-ku, Nagoya 466-8550, Japan. ✉email: kitagawa@kobepharma-u.ac.jp

Mammalian skin epithelium is a self-renewing tissue that constitutes the barrier between an organism and its environment. Epithelial keratinocyte proliferation is an essential element in wound healing, and abnormal epithelial proliferation is an intrinsic factor in the skin disorder psoriasis. The mechanisms that trigger epithelial proliferation are not completely understood. The inner basal layer adheres to an underlying basement membrane, which is rich in extracellular matrix components containing chondroitin sulfate (CS) proteoglycans. This layer contains proliferative keratinocytes that are typed by their expression of genes encoding keratins 5 and 14, and growth factor receptors such as the EGF receptor (EGFR)[1]. Here we have shown that the expression level of chondroitin 6-sulfate plays an important role in controlling the proliferation of basal keratinocytes via EGFR signaling.

Expression quantitative trait loci (eQTL) analysis shows that expression of the *FAM20B* gene is associated with psoriasis[2]. FAM20B is a kinase involved in xylose phosphorylation in the glycosaminoglycan (GAG)-protein linkage region[3]. FAM20B also regulates the number of GAG chains generated by a cell and plays an important role in the biosynthesis of GAG[3]. A previous study has demonstrated that *fam20b*-mutant zebrafishes have deficient production of cartilage CS proteoglycan and show chondrocyte hypertrophy in vivo[4]. In addition, *Fam20B*-knockout mouse embryos are severely stunted, showing multisystem organ hypoplasia, delayed development most evident in the skeletal system, eyes, lung, GI tract, and liver, and increased mortality at E13.5[5]. Koike et al. reported that FAM20B regulates the sulfation profile of CS chains and total amount of CS synthesized in cells[3]. Based on these findings, we hypothesized that a decrease in the expression of *FAM20B* would impact CS biosynthesis and would be associated with psoriasis.

CS chains consist of repeating disaccharide units $[(-4GlcUA\beta1-3GalNAc\beta1-)_n]$, and are covalently linked to specific serine (Ser) residues in any of the core proteins via GAG-protein linkage region (GlcUAβ1–3 Galβ1–3Galβ1–4Xylβ1–O-Ser)[6, 7]. The biosynthesis of the repeating disaccharide units is catalyzed by combined activity of six homologous glycosyltransferases – chondroitin (Chn) synthases (ChSy)−1, −2, and −3, Chn polymerizing factor (ChPF), and CS GalNAc transferases (CSGalNAcT)−1 and −2. The resulting backbones of CS chains are subsequently modified via sulfation and uronate epimerization[6]. Based on substrate preferences of Chn sulfotransferases identified to date, the biosynthetic scheme for CS-type sulfation is classified into the initial "4-*O*-sulfation" and "6-*O*-sulfation" pathways. In the initial step, the non-sulfated O unit [GlcUA-GalNAc] serves as a common acceptor substrate for two types of sulfotransferases, chondroitin 4-*O*-sulfotransferases (C4ST-1 and C4ST-2)[8–10] and chondroitin 6-*O*-sulfotransferase-1 (C6ST-1), forming monosulfated A (GlcUA-GalNAc(4-*O*-sulfate)) and C (GlcUA-GalNAc(6-*O*-sulfate)) units, respectively. Subsequent sulfation of A and C units can also occur via activity of GalNAc 4-sulfate 6-*O*-sulfotransferase (GalNAc4S-6ST) or CS-specific uronyl 2-*O*-sulfotransferase (UST), resulting in the formation of disulfated disaccharide E (GlcUA-GalNAc(4,6-*O*-disulfate)) and D (GlcUA(2-*O*-sulfate)-GalNAc(6-*O*-sulfate)) units, respectively[6]. Specific sulfation patterns of CS have been shown to affect cell proliferation. Chondroitin 6-sulfate, which has a high content of C units, is incorporated into collagen gels and promotes the growth of keratinocytes[11]. Chondroitin 4-sulfate is necessary for chondrocyte proliferation mediated by Indian hedgehog signaling[12]. Previous reports have shown that an imbalance in the sulfation of CS causes a hyperproliferative phenotype in cells derived from patients with Costello syndrome[13]. Notably, abnormal proliferation is a critical step in the pathogenesis of Costello syndrome, which is a combination of distinctive multiple congenital anomalies associated with increased cellular proliferation and tumor development. Costello syndrome is associated with severely reduced expression of C4ST-1 and loss of chondroitin 4-sulfate via constitutively activated HRAS signaling[13]. Chondroitin 6-sulfate is accumulated in the hearts of patients with Costello syndrome[14]. Restored expression of C4ST-1 can rescue increased proliferation of fibroblasts acquired from patients with Costello syndrome[13]. These findings indicate that the sulfation pattern of CS is a crucial regulator of proliferation, and that an imbalance in 6-*O*-sulfation and 4-*O*-sulfation of CS leads to pathological conditions.

*FAM20B*, a psoriasis-risk gene, regulates the sulfation profile of CS chains and total amount of GAG synthesized in cells. Increased expression levels of *FAM20B* are concomitant with increased proportion of CS 6-*O*-sulfation to 4-*O*-sulfation[3]. Therefore, we hypothesized that decreased expression of *FAM20B* may cause pathological conditions via decreases in the proportion of CS 6-*O*-sulfation to 4-*O*-sulfation. In this study, we used C6st-1 knockout mice that lack the expression of CS 6-sulfate to investigate how an imbalance in 6-*O*-sulfation to 4-*O*-sulfation affects CS and skin pathology. Here we show that *C6st-1* knockout mice exhibit keratinocyte hyperproliferation and impaired skin permeability barrier and that the expression level of chondroitin 6-sulfate plays an important role in controlling the proliferation of basal keratinocytes via EGFR signaling. This study suggests that the expression level of *C6st-1* may serve as a biomarker for the susceptibility to psoriasis.

## Results

**Percentage of 6-sulfated CS-disaccharides (C units) is decreased in psoriatic fibroblasts**. Figure 1a, b show the structure and biosynthesis of CS. Because psoriatic human epidermal keratinocytes are not commercially available, we first analyzed the CS produced by psoriatic human skin fibroblasts and normal human skin fibroblasts. The percentage of 6-sulfated CS-disaccharides (C units) in total CS-disaccharide units was slightly, but significantly, decreased in psoriatic fibroblasts (Fig. 1c).

***C6st-1* HE and KO mice show decreased expression of C and D units and epidermal hyperplasia**. Next, we analyzed mice lacking in CS 6-sulfation to examine whether a decrease in CS 6-sulfation would be associated with psoriasis. C6st-1 is involved in formation of 6-*O*-sulfated CS disaccharide units (C unit and D unit) (Fig. 1b). All mice produced similar amounts of CS, while production of the C and D units was decreased in *C6st-1* wild type (WT), *C6st-1* hetero (HE), and *C6st-1* knockout (KO) mice with respect to genotype (Fig. 1d, e, Supplementary Table 1). The expression levels of CS biosynthetic enzymes are shown in Fig. 1f. We examined the expression pattern of chondroitin 6-sulfate (CS-C) in the epidermis of WT mice (Fig. 1g). Immunohistochemical analysis using anti-CS-C antibody demonstrated that CS-C was expressed beneath the keratin 14 (K14)-positive stem cells of epidermal basal layer (Fig. 1g). In contrast, CS-C was not detected in the epidermal basal layer of *C6st-1* HE and *C6st-1* KO mice. Histological analyses revealed that in contrast with the thin epidermis of newborn and 8-week-old *C6st-1* WT mice, *C6st-1* HE, and *C6st-1* KO mice showed a hyperthickened epidermis (Fig. 1h, i). In addition, the epidermal thickness in newborn was thicker than that in 8-week-old mice. In this regard, it is reported that neonatal mice epidermal thickness in adult skin is thinner than that in infant skin because the epidermis reaches maximum thickness in the late embryo and cell proliferation gradually decreases within a few weeks postpartum[15]. Although there were differences in proliferation activity of keratinocytes between newborn and adult epidermis, C6st-1 affected both newborn and

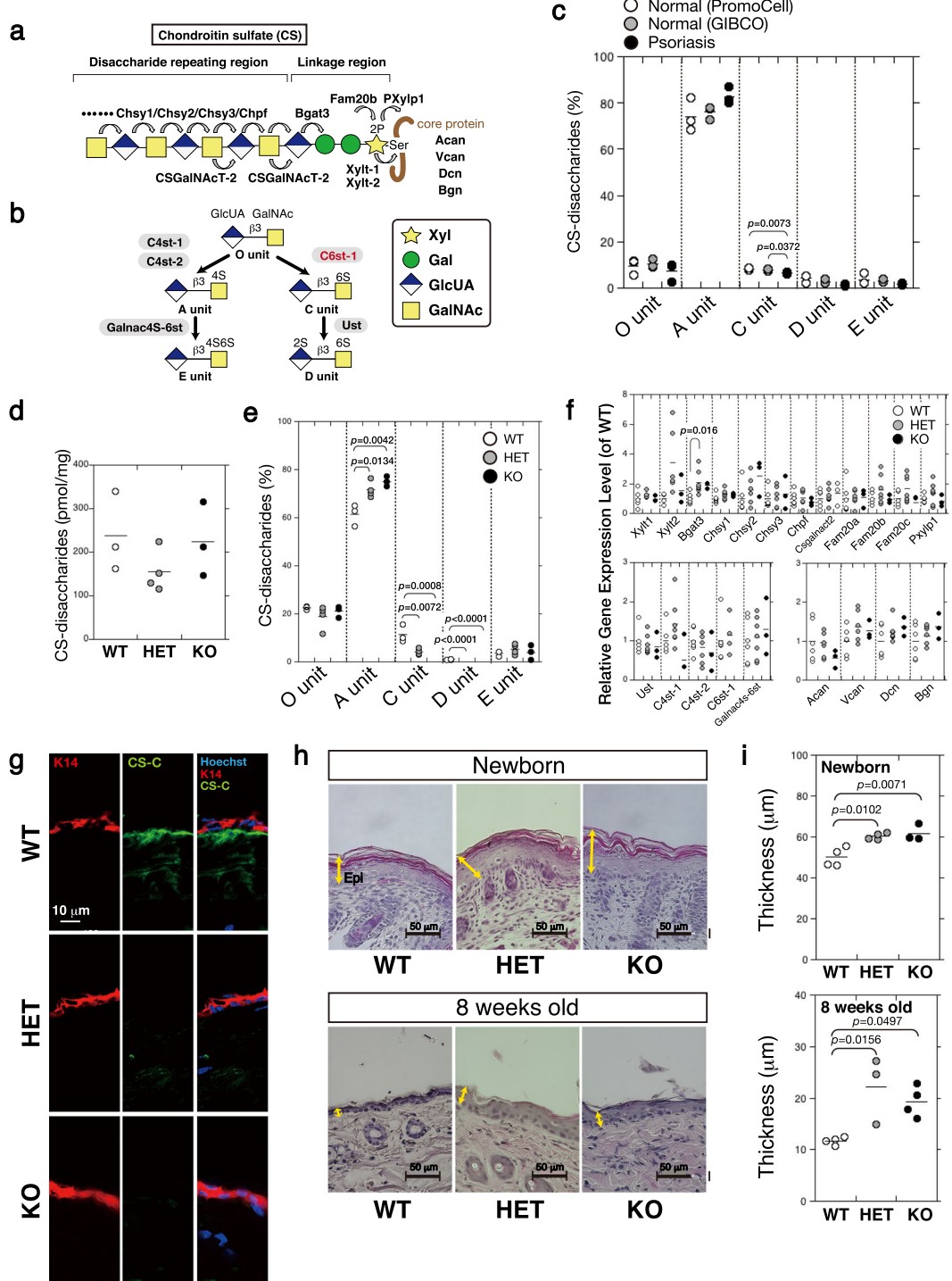

**Fig. 1 Downregulation of chondroitin 6-sulfate is associated with epidermal hyperplasia. a** The structure of CS chains is mediated by various glycosyltransferases. The common glycosaminoglycan–protein linkage region, GlcUAβ1–3Galβ1–3Galβ1–4Xylβ1-, is built on specific serine (Ser) residue(s) of core proteins such as ACAN, VCAN, DCN, and BGN. After the linkage region is formed, the CS polymerase complex assembles the CS backbone (disaccharide repeating region). **b** Outline of sulfation pathways. The C6-position of the GalNAc residue in the O unit is sulfated by C6ST-1 to form the C unit. Subsequently, the C unit is converted to a D unit by uronyl 2-O-sulfotransferase UST. **c** Expression of CS chains in normal human skin fibroblasts ($n = 3$) and psoriatic fibroblasts ($n = 3$) was analyzed using high pressure liquid chromatography (HPLC) to measure the composition of CS-disaccharides. Sulfated CS chains, isolated from the epidermis of *C6st-1* WT ($n = 3$), *C6st-1* HE ($n = 4$), and *C6st-1* KO mice ($n = 3$), were analyzed using HPLC to measure the total amount (**d**) and composition of CS-disaccharides (**e**). **f** Gene expression levels of CS biosynthetic enzymes in the epidermis of *C6st-1* WT ($n = 4$-6), *C6st-1* HE ($n = 4$-7), and *C6st-1* KO mice ($n = 3$) were analyzed using real-time PCR. **g** The expression pattern of 6-O-sulfated CS (CS-C) in the tail epidermis of *C6st-1* WT, *C6st-1* HE, and *C6st-1* KO adult mice was examined by immunohistochemical analysis using anti-keratin 14 (K14) and anti-CS-C antibody. **h** Hematoxylin-eosin staining of paraffin-embedded skin samples from newborn and 8-week-old *C6st-1* WT, *C6st-1* HE, and *C6st-1* KO mouse skin. Yellow arrows indicate epidermis. **i** Epidermal thickness of newborn and 8-week-old *C6st-1* WT, *C6st-1* HE, and *C6st-1* KO mice. Newborn mice, *C6st-1* WT ($n = 4$), *C6st-1* HE ($n = 4$), and *C6st-1* KO ($n = 3$); 8-week-old mice, *C6st-1* WT ($n = 4$), *C6st-1* HE ($n = 3$), and *C6st-1* KO ($n = 4$). Statistical significance was determined using one-way ANOVA with Tukey's HSD test.

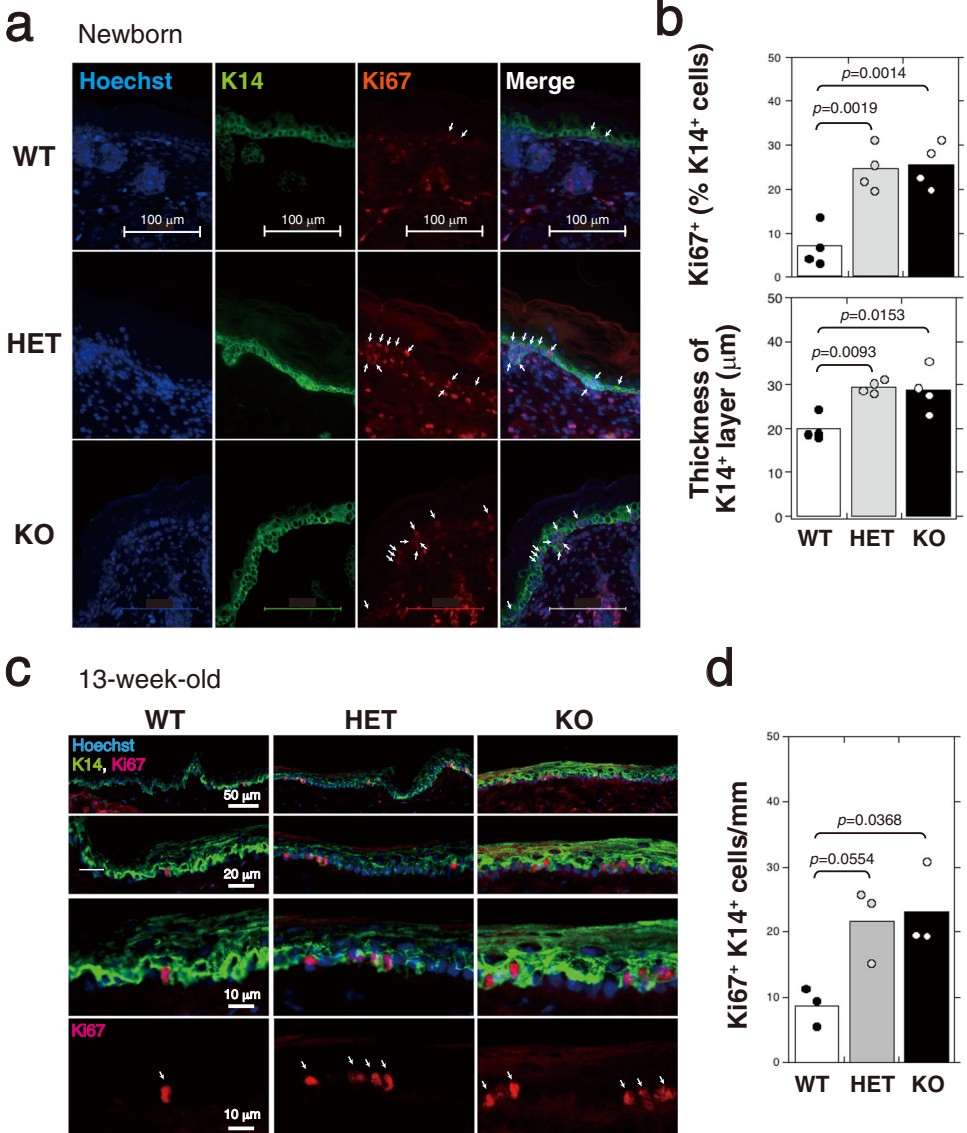

**Fig. 2 Proliferation of keratinocytes is accelerated by downregulation of CS-C. a** Immunofluorescence labeling of skin from newborn *C6st-1* WT, *C6st-1* HE, and *C6st-1* KO mice was performed using anti-K14 and anti-Ki67 antibodies. Nuclei are counterstained with Hoechst33342. **b** Quantification of Ki67-positive cells in K14-positive basal layers, and measurement of epidermal thickness in K14-positive basal layer of *C6st-1* WT ($n = 4$), *C6st-1* HE ($n = 4$), and *C6st-1* KO mice ($n = 4$). Multiple random vertical lines perpendicular to the epidermal border were measured. From the mean thickness of these lines, epidermal thickness in K14-positive basal layer was calculated. **c** Immunofluorescence labeling of tail skin from 13-week-old *C6st-1* WT, *C6st-1* HE, and *C6st-1* KO mice was performed using anti-K14 and anti-Ki67 antibodies. Nuclei are counterstained with Hoechst33342. **d** Numbers of Ki67-positive cells within 1 mm of K14-positive basal layers in *C6st-1* WT ($n = 3$), *C6st-1* HE ($n = 3$), and *C6st-1* KO mice ($n = 3$). The number of Ki67-positive cells per mm of basal layer length was counted manually using a digital imaging software (Photoshop CS6). Statistical significance was determined using one-way ANOVA with Tukey's HSD test.

adult epidermis. Notably, there was a possibility that a decrease by half in the expression level of *C6st-1* could affect K14-positive basal cells to cause epidermal hyperplasia (HET in Fig. 1h, i).

**Proliferating K14-positive basal keratinocytes is increased in newborn and 13-week-old HE and KO mice.** We next examined whether epidermal proliferation and differentiation are affected by *C6st-1* ablation. *C6st-1*-null epidermis appeared morphologically normal; the expression patterns of basal keratin 14 (K14), which is a marker protein for non-differentiated and proliferative keratinocytes, were indistinguishable from those in *C6st-1* WT skin (Fig. 2). Proliferating K14-positive basal keratinocytes were examined by immunolabeling for Ki67, which is expressed in the nucleus of cycling cells. The number of proliferating K14-positive

basal keratinocytes was increased in newborn and 13-week-old *C6st-1* HE and *C6st*-1 KO mice compared with that in *C6st-1* WT mice (Fig. 2). Consistent with this result, the K14-positive cells in the basal layer were mitotically more active in newborn *C6st*-1 HE and *C6st*-1 KO epidermis than in *C6st-1* WT epidermis (Fig. 2b). These results suggest that proliferation is enhanced by loss of *C6st*-1 in the basal layer.

**Loss of *C6st-1* affects keratinocyte differentiation.** We next examined whether loss of *C6st-1* affected keratinocyte differentiation. In normal epidermis, K14-positive basal keratinocytes exit from mitosis and differentiate into keratins 10 (K10)-positive spinous keratinocytes[16]. Although the expression of K10 was associated with the loss of proliferative capacity in the *C6st-1* WT

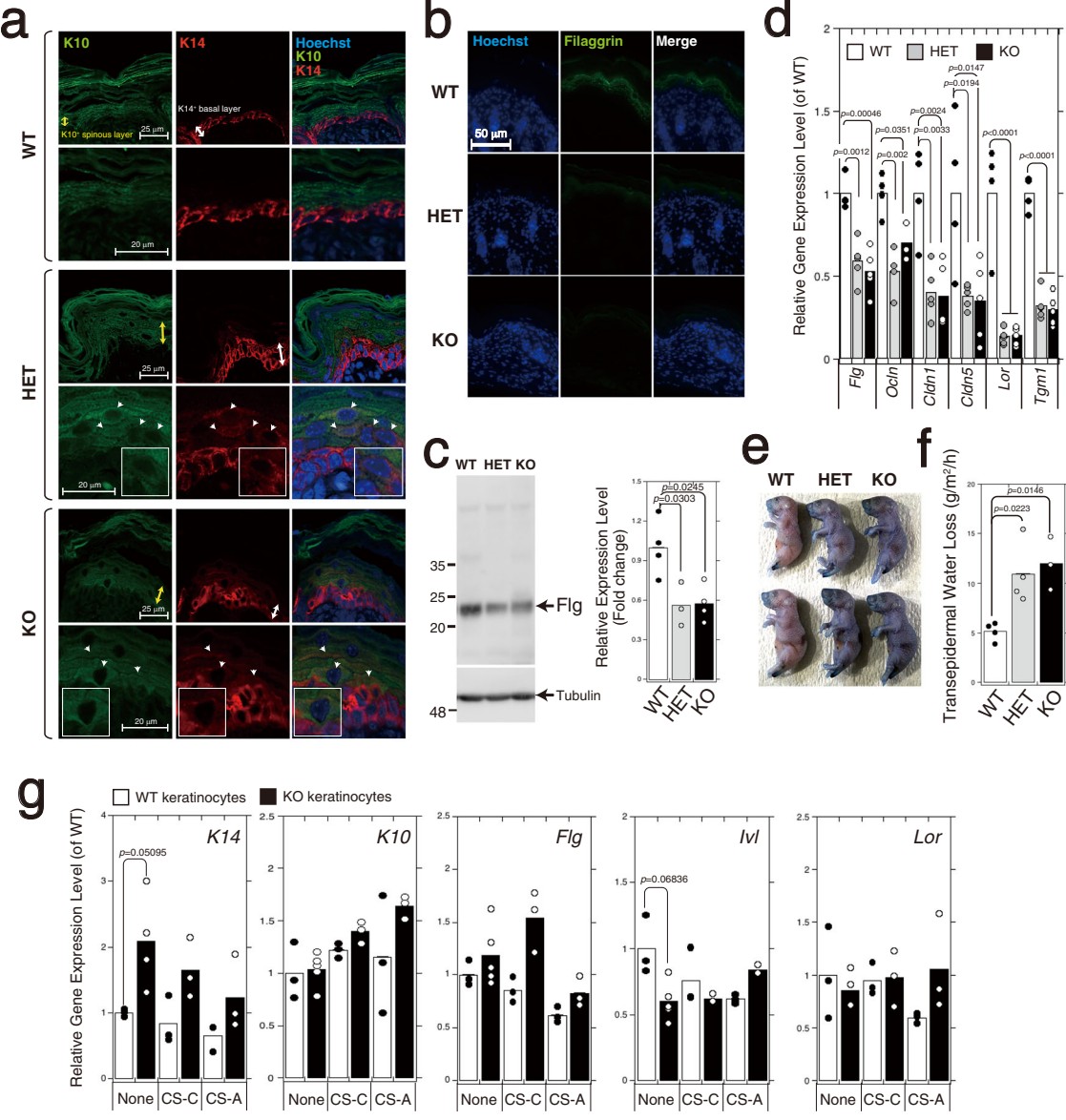

**Fig. 3 Expression of skin proliferation/differentiation markers and skin barrier function is disrupted in _C6st-1_ HE and _C6st-1_ KO mice. a**
Immunofluorescence analysis of skin sections from _C6st-1_ WT, _C6st-1_ HE, and _C6st-1_ KO mice is presented. Green signals indicate K10 and red signals show K14. Positive labeling for K10 and K14 in the spinous layer (yellow arrows) and basal layer (white arrows), respectively, is shown. Magnified images are shown below. Arrowheads indicate K10- and K14-double positive cells, and one of them is shown in the inset. **b** Immunofluorescence labeling of sections from the back skin of _C6st-1_ WT, _C6st-1_ HE, and _C6st-1_ KO mice was conducted using an anti-filaggrin antibody (green). Nuclei were counterstained with Hoechst (blue). **c** Protein extracts from epidermis of _C6st-1_ WT ($n = 4$), _C6st-1_ HE ($n = 3$), and _C6st-1_ KO ($n = 4$) were immunoblotted with anti-filaggrin and anti-tubulin antibodies. Tubulin was used as loading control. Statistical significance was determined using one-way ANOVA with Tukey's HSD test. **d** The expression levels of filaggrin (_Flg_), occludin (_Ocln_), claudin 1 (_Cldn1_), claudin 5 (_Cldn5_), loricrin (_Lor_), and transglutaminase 1 (_Tgm1_) in samples of _C6st-1_ WT ($n = 4$), _C6st-1_ HE ($n = 4$-5), and _C6st-1_ KO epidermis ($n = 3$-5) were analyzed using real-time PCR. Statistical significance was determined using one-way ANOVA with Tukey's HSD test. **e** Euthanized neonatal pups were immersed in toluidine blue solution. **f** TEWL was determined in 8-week-old _C6st-1_ WT ($n = 4$), _C6st-1_ HE ($n = 4$), and _C6st-1_ KO mice ($n = 3$). Statistical significance was determined using one-way ANOVA with Tukey's HSD test. **g** Expression levels of differentiation markers (_K14, K10, Flg, Ivl,_ and _Lor_) in _C6st-1_ WT and _C6st-1_ KO keratinocytes in the presence or absence of CS-C or CS-A were analyzed using real-time PCR ($n = 3$-5). Values represent the mean ± standard deviation. Statistical significance was determined using Student's _t_ test.

epidermis, double-positive K14 and K10 keratinocytes were detected in the spinous layer of _C6st_-1 HE and _C6st_-1 KO mice (Fig. 3a, indicated by arrowheads). Filaggrin is an intermediate filament-associated protein that aids in the packing of keratin filaments and terminal differentiation processes of keratinocytes by facilitating apoptotic machinery[17]. The expression levels of filaggrin in the granular layer of _C6st_-1 HE and _C6st_-1 KO mice

were significantly lower compared with those in _C6st_-1 WT mice (Fig. 3b, c and Supplementary Fig. 1). Furthermore, proteins involved in cellular junctions and barrier formation are produced by differentiated cells such as spinous and granular cells. The gene expression levels of filaggrin, occludin, claudin 1, claudin 5, loricrin, and transglutaminase 1 were significantly decreased in _C6st_-1 HE and _C6st_-1 KO mice compared with those in _C6st_-1

WT mice (Fig. 3d). These results suggest that loss of C6st-1 affects epidermal differentiation and epidermal proliferation. We next examined the gene expression levels of differentiation markers (K14, K10, Flg, Ivl, and Lor) in primary keratinocytes (Fig. 3g). The expression level of K14 was elevated in C6st-1 KO keratinocytes compared with C6st-1 WT keratinocytes, whereas Ivl was down-regulated in C6st-1 KO keratinocytes. However, altered differentiation marker expression levels could not be rescued by addition of CS-C to C6st-1 KO keratinocytes. There were no significant differences in the expression levels of K10, Flg, and Lor between C6st-1 WT and C6st-1 KO keratinocytes (Fig. 3g). Thus, gene expression levels of differentiation markers may be low during in vitro culture even in C6st-1 WT keratinocytes. These results suggest that differentiation status of C6st-1 KO keratinocytes is slightly altered by loss of CS-C. However, it is thought that CS-C does not directly regulate keratinocyte differentiation.

**Epidermal permeability barrier function is impaired by loss of C6st-1.** Loss of filaggrin impacts the skin barrier function[18]. Decreased levels of proteins required for formation of tight junctions and the cornified envelope can also disrupt the skin barrier function. Thus, we performed a dye penetration assay using toluidine blue dye on newborn C6st-1 WT, C6st-1 HE, and C6st-1 KO pups (Fig. 3e). C6st-1 HE and C6st-1 KO pups showed increased dye penetration compared with that observed in C6st-1 WT pups (Fig. 3e). Transepidermal water loss (TEWL) measurements indicated that C6st-1 HE and C6st-1 KO mice showed a significant increase in TEWL values compared with those of C6st-1 WT mice (Fig. 3f). These results suggest that both the "outside-in" and "inside-out" functions of the epidermal permeability barrier were impaired by loss of C6st-1.

**Ablation of C6st-1 affects proliferative and inflammatory signaling in the epidermis.** We hypothesized that the loss of C6st-1 can cause an abnormality in the proliferative signaling of the epidermis. EGFR signaling is particularly important in regulating proliferation in the epidermis[19]. Phosphorylation of EGFR was significantly elevated in C6st-1 HE and C6st-1 KO epidermis compared with that in C6st-1 WT epidermis (Fig. 4a, b, and Supplementary Fig. 1). Stimulation of EGFR typically leads to downstream activation of ERK1/2 and STAT3[1, 20]. ERK1/2 and STAT3 were activated in C6st-1 HE and C6st-1 KO epidermis at a higher level than in the C6st-1 WT epidermis (Fig. 4a, b, and Supplementary Fig. 1). Additionally, the major ligands for EGFR in C6st-1 HE and C6st-1 KO mice are expressed at levels similar to those in C6st-1 WT mice (Fig. 4c). Furthermore, the gene expression levels of several pro-inflammatory cytokines, associated with STAT3 activation, were examined (Fig. 4d). Expression levels of Il1b, Il6, Il7, and Il23 were unaffected by C6st-1 expression level in the dorsal skin of newborn mice (Fig. 4d). In adult mouse skin, expression levels of Il6, Il7, and Il23 were elevated by loss of C6st-1 (Fig. 4d). These results suggest that ablation of C6st-1 does not directly activate intrinsic pro-inflammatory signaling but may impair epidermal barrier function, producing a state more primed for inflammation.

**Keratinocyte proliferation is cell-autonomously stimulated in an EGFR-dependent manner in the absence of C6st-1.** We next examined whether EGFR signaling was cell-autonomously activated in C6st-1 HE and C6st-1 KO keratinocytes. As shown in Supplementary Table 2, 7 mol% of C units were contained in CS chains produced by primary keratinocytes. The expression levels of phosphorylated EGFR and phosphorylated STAT3 were significantly elevated in primary keratinocytes derived from C6st-1 HE and C6st-1 KO mice compared with the levels in keratinocytes derived from C6st-1 WT mice (Fig. 5b, c, and Supplementary Fig. 1). Furthermore, C6st-1 HE and C6st-1 KO keratinocytes showed a highly proliferative phenotype compared with that of C6st-1 WT keratinocytes (Fig. 5a). PD153035, a specific inhibitor of EGFR, effectively blocked the phosphorylation of EGFR and STAT3 (Fig. 5b, c, and Supplementary Fig. 1), and proliferation of C6st-1 KO keratinocytes was lowered to the same level as that of C6st-1 WT keratinocytes (Fig. 5a). Notably, the expression levels of phosphorylated EGFR and phosphorylated STAT3 were returned to WT levels by exogenously added CS-C but not by exogenously added chondroitin 4-sulfates (CS-A) (Fig. 5b, c, and Supplementary Fig. 1). Consistent with these results, the hyperproliferative phenotype of C6st-1 KO keratinocytes was recovered after treatment with CS-C (Fig. 5a). These results indicate that proliferation of keratinocytes is cell-autonomously stimulated in an EGFR-dependent manner in the absence of C6st-1.

**CS-C regulates EGFR signaling.** We investigated the mechanism underlying the regulation of EGFR signaling by CS-C. A direct interaction between EGFR and CS-C was visualized and quantified using proximity ligation assay (Fig. 6a, c). Signals indicating direct interactions between EGFR with chondroitin 6-sulfate (EGFR-CS) were detected in C6st-1 WT keratinocytes (Fig. 6a). These signals were diminished after C6st-1 WT keratinocytes were pre-treated with chondroitinase ABC (Chase ABC) (Fig. 6a, c). EGFR-CS signals were decreased in C6st-1 HE and C6st-1 KO keratinocytes (Fig. 6a, c); however, EGFR homodimers in C6st-1 HE and C6st-1 KO keratinocytes were expressed at levels similar to those in C6st-1 WT keratinocytes (Fig. 6b, d). These results suggest that chondroitin 6-sulfates directly bind to EGFR and block EGFR signaling.

Furthermore, we investigated direct interaction of CS-C with EGFR (binding of these proteins to each other) using surface plasmon resonance (SPR; Fig. 6e). While CS-C could bind to EGFR with high affinity, CS-A could not. In addition, we investigated whether CS-C could interact with EGF ligands (Fig. 6f). Neither CS-C nor CS-A bound to EGF.

**CS-C regulates human keratinocytes via EGFR signaling.** We confirmed that CS-C regulates the proliferation of human keratinocytes via EGFR signaling[21] (Fig. 7). CS chains produced in two human keratinocyte clones, HaCaT and PSVK1, contained C units (Supplementary Table 2). Depletion of CS chains by digestion with Chase ABC accelerated the proliferation of two human keratinocyte clones, HaCaT and PSVK1 (Fig. 7d, e) with concomitant activation of EGFR (Fig. 7b, c, and Supplementary Fig. 1). In addition, Chase ABC-mediated enhancement of proliferation could be rescued by treatment with the EGFR inhibitor PD153035 (Fig. 7d, e). Furthermore, interactions between CS-C and EGFR were detected in both HaCaT and PSVK1 cells (Fig. 7f, g).

**Imiquimod (IMQ)-treated C6st-1 HE and C6st-1 KO mice are more vulnerable to treatment with IMQ than are C6st-1 WT mice.** Psoriasis did not develop as a result of enhanced proliferation of keratinocytes induced by lowered expression of CS-C. We used the imiquimod (IMQ)-induced psoriasis model in WT, C6st-1 HE, and C6st-1 KO mice to investigate the role of CS-C in the epidermis under conditions of inflammation. For this, we swabbed IMQ-containing cream daily for 4 days onto the shaved backs of 6-week-old WT, C6st-1 HE, and C6st-1 KO mice; skin was obtained for analysis on day 3. As shown in Fig. 8, C6st-1 HE and C6st-1 KO mice showed increased levels of acute skin inflammation (manifested by erythema, scaling, and skin induration) and thicker

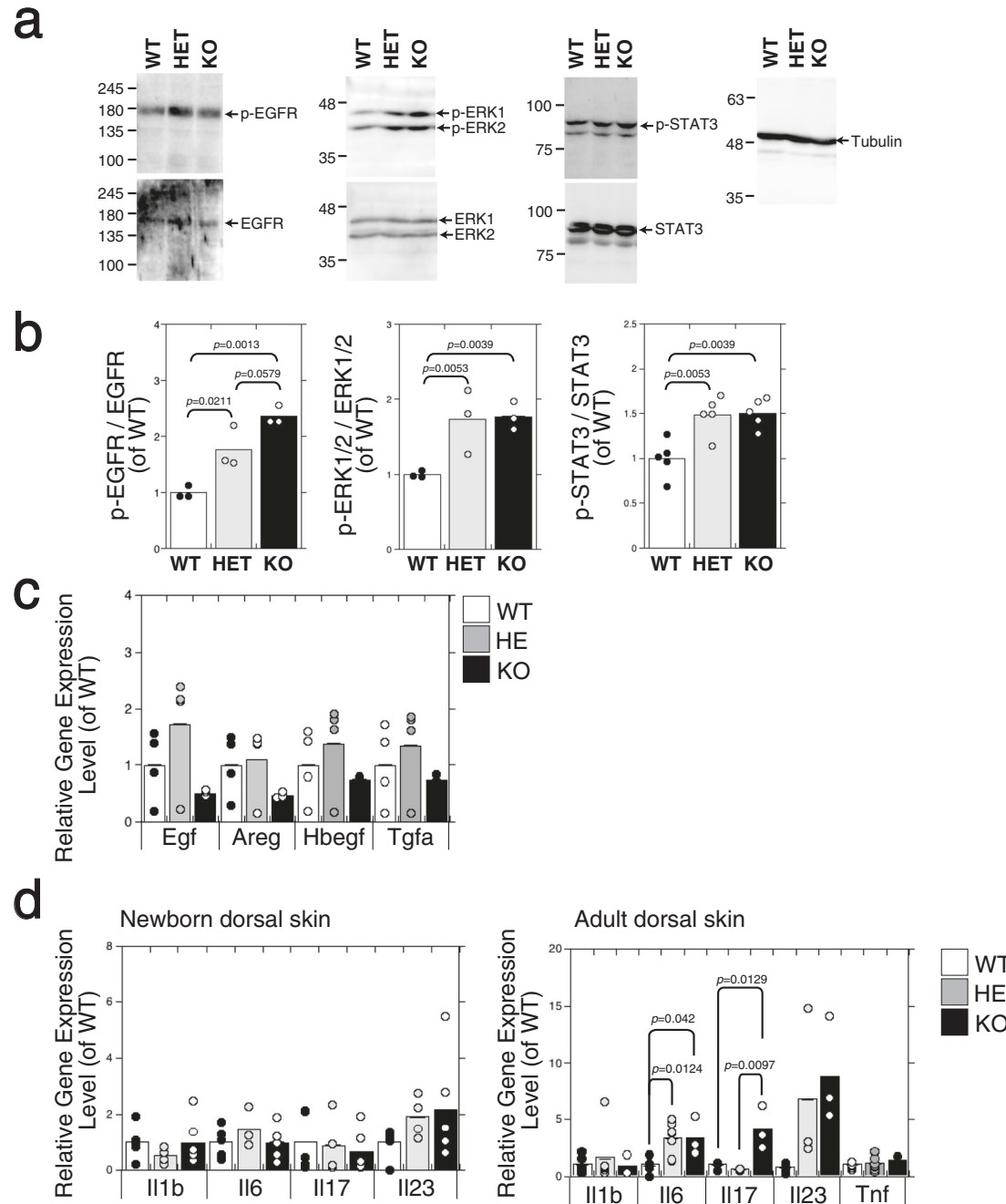

**Fig. 4 The expression of phospho-EGFR, phospho-ERK1/2, and phospho-STAT3 is augmented in the epidermis of *C6st-1* HE and *C6st-1* KO mice compared with that in *C6st-1* WT mice. a** Proteins extracted from the epidermis of newborn *C6st-1* WT, *C6st-1* HE, and *C6st-1* KO mice were analyzed by immunoblotting using anti-phospho-EGFR (p-EGFR), anti-EGFR, anti-phospho-ERK1/2 (p-ERK1/2), anti-ERK1/2, and anti-phospho-STAT3 (p-STAT3), anti-STAT3, and anti-tubulin antibodies. **b** Densitometric analysis of the levels of phospho-EGFR, phospho-ERK1/2, and phospho-STAT3 compared with levels of total EGFR, ERK1/2, and STAT3 is shown. Bars represent the means ± S.D. from three independent biological replicates. Statistical significance was determined using one-way ANOVA with Tukey's HSD test. **c** The expression levels of EGFR ligands (*Egf, Areg, Hbegf, and Tgfa*) in *C6st-1* WT, *C6st-1* HE, and *C6st-1* KO keratinocytes were analyzed using real-time PCR ($n = 4$). Values are mean ± S.D. **d** Expression levels of pro-inflammatory cytokines (*Il1b, Il6, Il17,* and *Il23*) in dorsal skin of *C6st-1* WT, *C6st-1* HE, and *C6st-1* KO newborn and adult (13-week-old) mice were analyzed using real-time PCR ($n = 3–7$). Values represent the mean ± S.D. Statistical significance was determined using one-way ANOVA with Tukey's HSD test.

epidermis at day 3 of IMQ treatment compared with those of WT mice (Fig. 8a, b, c). Psoriasis is characterized by dysregulated proliferation and differentiation of keratinocytes. K14 and Ki67 double-positive keratinocytes were increased in basal layer of WT, *C6st-1* HE, and *C6st-1* KO mice at 1 day after the first application of IMQ (Fig. 8d, e). Under IMQ-induced inflammation, keratinocyte

proliferation showed increased activation in *C6st-1* HE and *C6st-1* KO mice compared with that in *C6st-1* WT mice (Fig. 8e). After WT, *C6st-1* HE, and *C6st-1* KO mice were treated with IMQ for 3 days, the K14-positive basal layer was markedly thicker in *C6st-1* HE and *C6st-1* KO mice than in WT mice (Fig. 8f, g). In addition, the levels of phosphorylated EGFR were elevated in *C6st-1* HE and

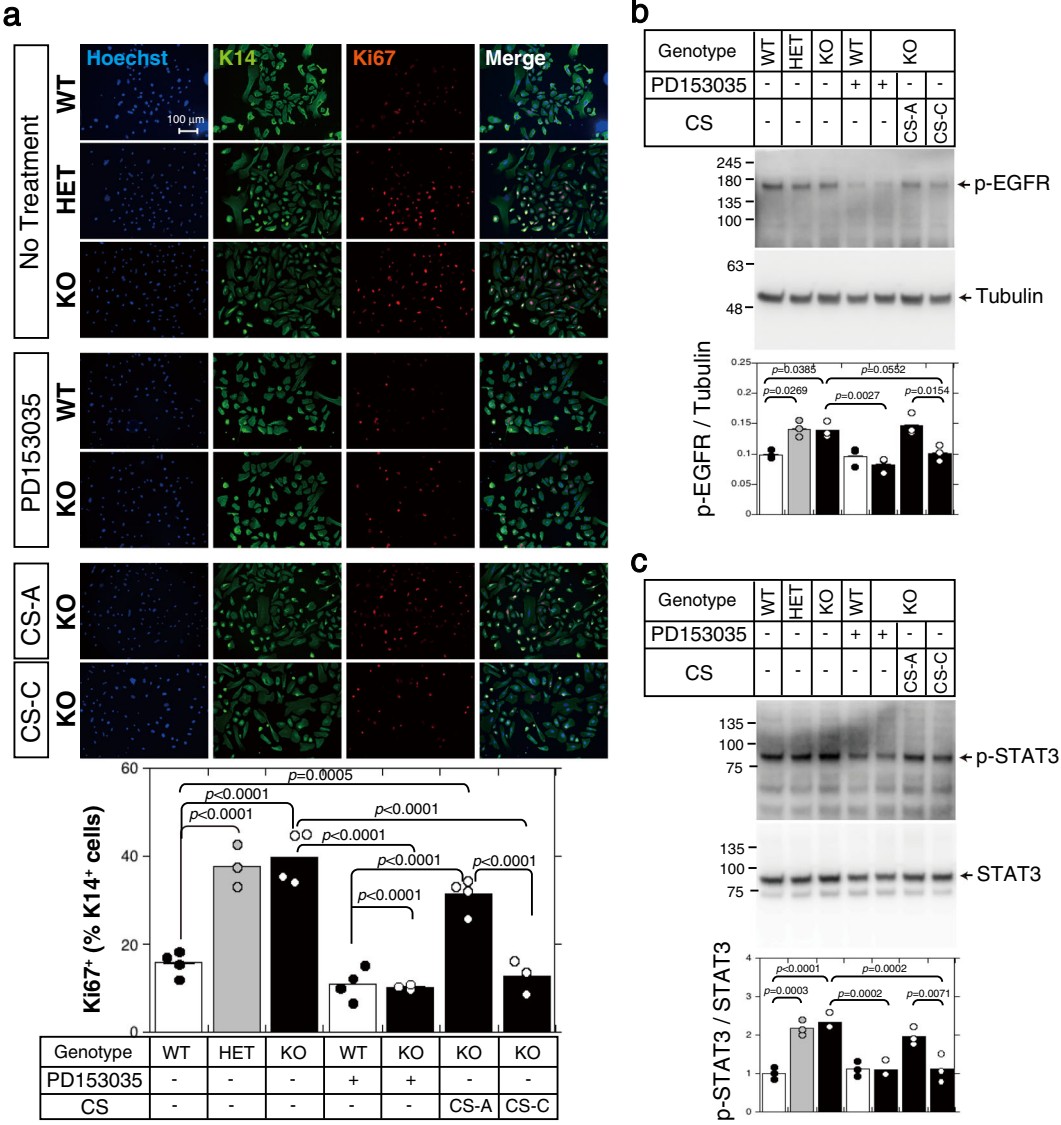

**Fig. 5 EGFR-dependent keratinocyte proliferation is regulated by CS-C. a** Keratinocytes ($1 \times 10^5$ cells/well), isolated from *C6st-1* WT, *C6st-1* HE, and *C6st-1* KO epidermis, were cultured in the presence or absence of 5 μM PD153035, 100 μg/ml CS-A, and 100 μg/ml CS-C. Cells were immunolabeled using anti-K14 and anti-Ki67 antibodies. The graph below indicates quantification of Ki67- and K14-double positive cells. Values are mean ± S.D. with $n = 3$-4 per group. Statistical significance was determined using one-way ANOVA with Tukey's HSD test. Keratinocytes were treated with 5 μM PD153035, 100 μg/ml CS-A, and 100 μg/ml CS-C as indicated. Cells were lysed and subjected to immunoblotting using **b** anti-phospho-EGFR and anti-tubulin antibodies, and **c** anti-phospho-STAT3 and anti-STAT3 antibodies. Tubulin and STAT3 were used as loading controls. The graphs below the blots show relative expression levels of phospho-EGFR and phosphor STAT3 in *C6st-1* WT, *C6st-1* HE, and *C6st-1* KO keratinocytes subjected to the indicated treatments. Values are mean ± S.D. with $n = 3$ per group. Statistical significance was determined using one-way ANOVA with Tukey's HSD test.

KO epidermis at 8 h after application of IMQ compared with those in untreated epidermis; EGFR was not activated in *C6st-1* WT epidermis (Fig. 8h and Supplementary Fig. 1). Furthermore, the number of K14 and Ki67 double-positive keratinocytes in *C6st-1* HE and *C6st-1* KO mice was higher than that in *C6st-1* WT mice at 8, 24, and 96 h after application of IMQ (Fig. 8i). IMQ-treated skin in *C6st-1* HE and *C6st-1* KO mice showed increased epidermal thickening as early as 24 h after application of IMQ, and became further thicker between 24 and 72 h post application of IMQ (Fig. 8j). In contrast, *C6st-1* WT epidermis increased in thickness at 72 h after application of IMQ, and became much thicker between 72 and 120 h post application (Fig. 8j). These results indicate that skin hypertrophy in the IMQ-treated *C6st-1* HE and *C6st-1* KO mice was accelerated compared with *C6st-1* WT mice.

## Discussion

Proliferation of epidermal keratinocytes is tightly controlled under normal physiological conditions. In many common skin diseases, such as in psoriasis, inflammation disrupts the controlled proliferation of keratinocytes and causes pathological epidermal hyperplasia[22–24]. Numerous studies have shown that abnormal proliferation of keratinocytes is triggered by inflammation under pathological conditions. In this study, we examined whether CS, expressed in the extracellular matrix and on cell surface, control proliferation of keratinocytes under normal conditions.

The proliferative EGFR-expressing keratinocytes reside in the basal layer, which adheres to an underlying basement membrane that is rich in 6-sulfated CS (Fig. 1g). The expression of

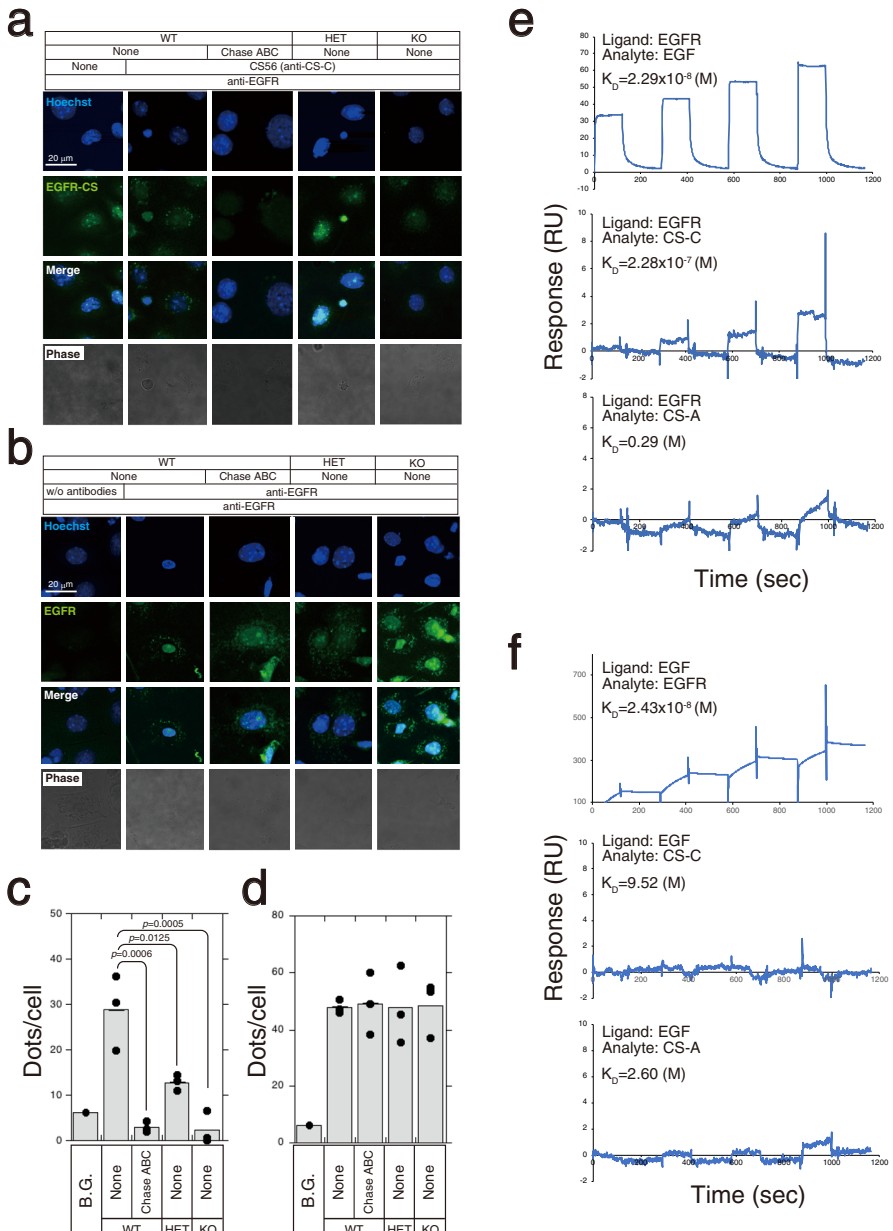

**Fig. 6 CS-C directly interacts with EGFR in mouse keratinocytes. a** Direct interactions between EGFR and CS-C (EGFR-CS) in primary *C6st-1* WT, *C6st-1* HE, and *C6st-1* KO keratinocytes were examined by proximity ligation assay (PLA assay) using CS56 (anti-CS-C antibody) and anti-EGFR antibody. Keratinocytes digested with chondroitinase ABC were used as controls. **b** EGFR homodimers in primary keratinocytes were examined by PLA assay using anti-EGFR antibody. As a control, keratinocytes were digested with chondroitinase ABC, subjected to PLA assay. **c** The graph shows means ± S.D. of EGFR-CS signals (green dots)/cell counted in 7–12 cells from three independent experiments. **d** The graph shows means ± S.D. of EGFR homodimer signals (green dots)/cell counted in 7–12 cells from three independent experiments. Values are mean ± S.D. Statistical significance was determined using one-way ANOVA with Tukey's HSD test. **e** The interaction of EGFR with EGF, CS-C, or CS-A was analyzed via SPR. **f** Similarly, the interaction of EGF with EGFR, CS-C, or CS-A was analyzed by SPR.

chondroitin 6-sulfate (CS-C) in the basement membrane has previously been shown by immunohistochemical analysis using a murine monoclonal antibody against CS-C[25]. Therefore, we analyzed the epidermis of mice lacking *C6st-1*, which encodes the sulfotransferase involved in biosynthesis of CS-C. Loss of CS-C caused keratinocyte hyperproliferation mediated by hyperactivation of EGFR (Figs. 2 and 4). Keratinocyte proliferation is regulated in a cell-autonomous manner under normal conditions because CS-C produced by basal keratinocytes negatively controls the activation of EGFR (Fig. 5). Normal levels of keratinocyte

hyperproliferation were reestablished by exogenously added CS-C (Fig. 5). In addition, hyperproliferation caused by loss of CS-C may indirectly disturb keratinocyte differentiation and thus epidermal barrier function (Fig. 3). Such a sequence of events initiated by loss of CS-C may make keratinocytes more vulnerable to environmental stress (Fig. 9). Thus, we consider that *C6st-1*KO keratinocytes exist in a primed activation state following inflammatory stimuli. A primed activation state may not necessarily involve increased expression of pro-inflammatory cytokines. This primed state can be conceptualized as "readiness"

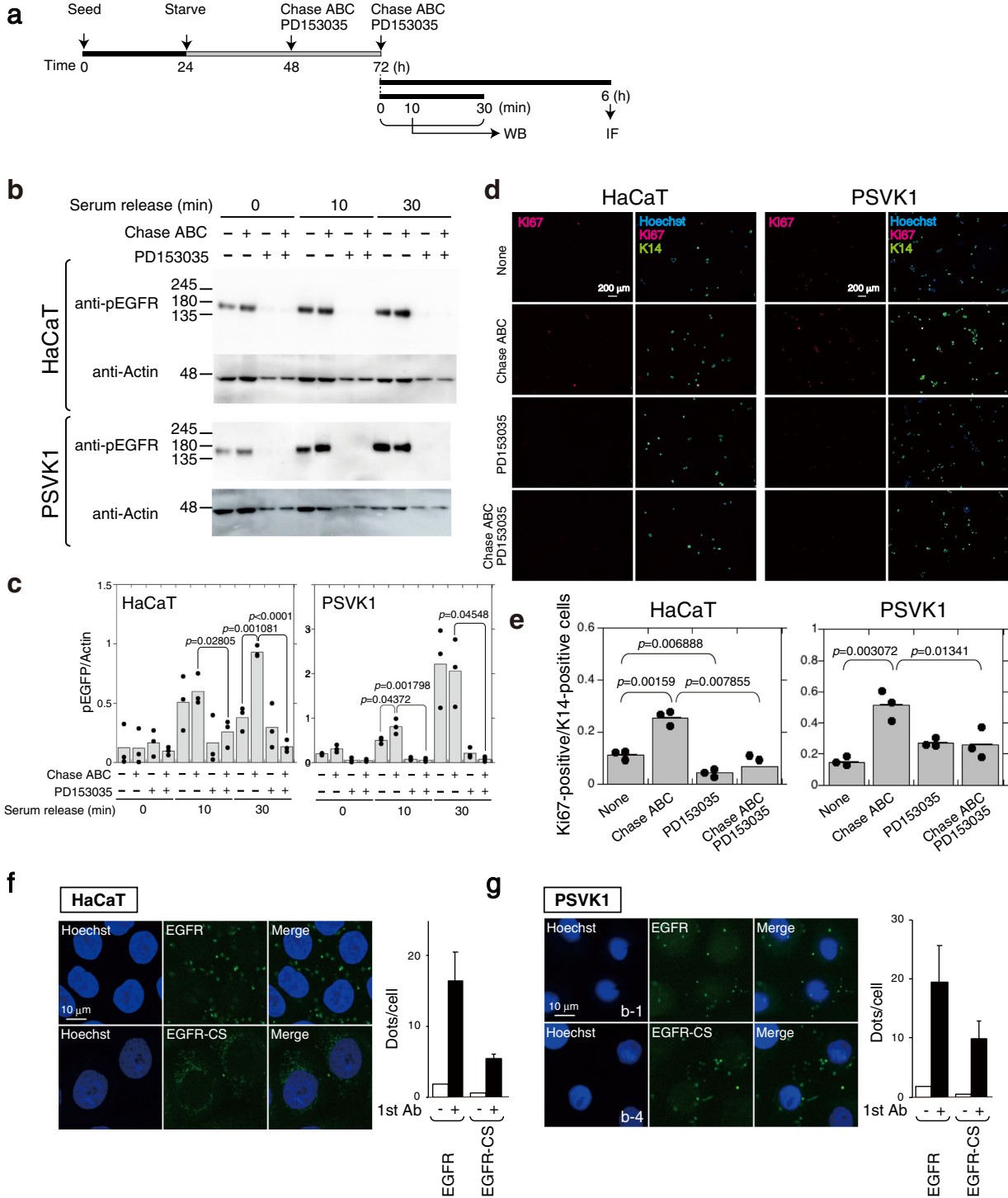

**Fig. 7 Proliferation of cells from human keratinocyte clone lines, HaCaT and PSVK1, is regulated by CS. a** Schematic diagram of experimental workflow. **b** HaCaT cells ($2 \times 10^4$ cells/dish) and PSVK1 cells ($1.7 \times 10^5$ cells/dish) were seeded, and starved for 48 h prior to allowing cell cycle progression via addition of serum in the presence of 2.5 munits/mL of Chase ABC and 5 μM PD153035, and culturing cycling cells for the indicated time. Proteins extracted from HaCaT and PSVK1 cells were analyzed by immunoblotting using anti-phospho-EGFR (p-EGFR), and anti-EGFR antibody. **c** Densitometric analysis of the levels of phospho-EGFR compared with levels of actin is shown. Bars represent the means ± S.D. from three independent biological replicates. Statistical significance of differences was determined using Student's *t* test. **d** HaCaT ($1 \times 10^4$ cells/well) and PSVK1 cells ($2.5 \times 10^4$ cells/well) were cultured in the presence or absence of Chase ABC (2.5 munits) and PD153035 (5 μM), and stained with anti-K14 and anti-Ki67 antibody. Nuclei were counterstained with Hoechst33342. **e** Quantification of Ki67- and K14-double positive keratinocytes is shown. Values are mean ± S.D. from three independent biological replicates. Statistical significance was determined using Student's *t* test. Direct interactions between EGFR and CS-C (EGFR-CS) and EGFR homodimers in HaCaT (**f**) and PSVK1 (**g**) cells were detected by PLA assay. The graph shows means ± S.D. of EGFR homodimer and EGFR-CS signals (green dots)/cell counted from three independent experiments.

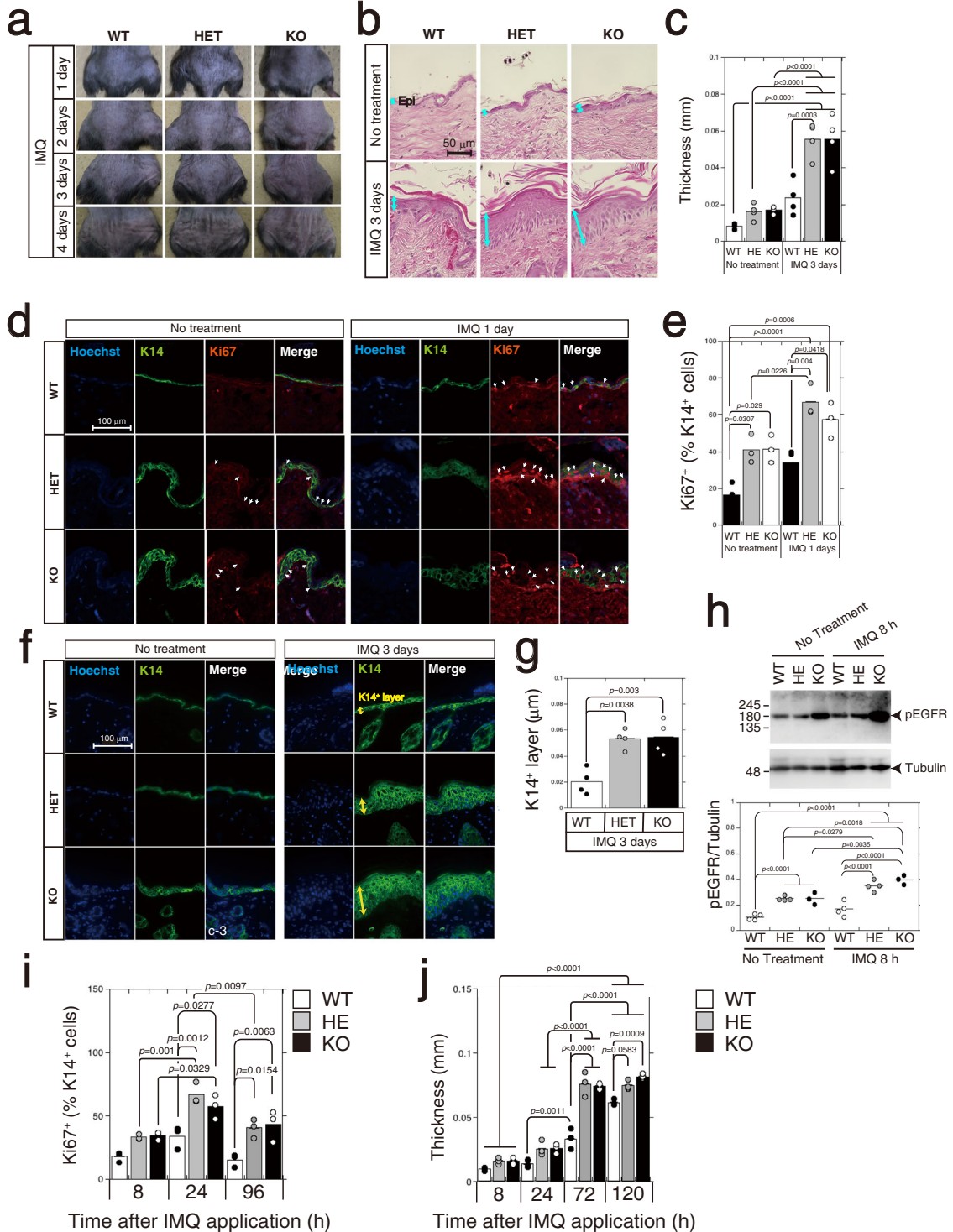

state. Thus, *C6st-1*KO keratinocytes may rapidly respond or have a low-threshold response to inflammatory stimuli.

The mitotically active basal layer of the epidermis contains stem cells (basal cells). The basal layer produces and secretes extracellular matrix constituents, including CS-C, which contributes to stem cell niche maintenance. As shown, CS-C localizes primarily to the basement membrane (Fig. 1g). Contact of epidermal stem cells with basement membrane is of fundamental importance in regulating their proliferation and differentiation. Thus, the present study identifies CS-C as a regulatory stem cell niche component within skin.

One manifestation of aging is the decline in epidermal barrier function. In human skin, CS-C is specifically expressed in the basal lamina, and decreases in an age-dependent manner[26]. Results of the present study suggest that a decrease in CS-C impairs barrier function via dysregulating keratinocyte proliferation. Thus, age-related changes in CS-C may be causally associated with functional changes intrinsic to the aging process of human skin.

CS-C, hyaluronic acid, and collagens serve as substrates supporting the normal functioning of human keratinocytes and fibroblasts. These biomaterials can promote wound healing[27]. Although biomaterials based on CS-C can serve as scaffolds for

**Fig. 8 CS-C regulates keratinocyte proliferation and skin thickness in imiquimod-induced inflammation model of psoriasis. a** Macroscopic phenotypical representation of psoriatic lesions in *C6st-1* WT, *C6st-1* HE, and *C6st-1* KO mice, treated daily with imiquimod. Images acquired at 1, 2, 3, and 4 days after initial treatment. **b** Back skin sections from *C6st-1* WT, *C6st-1* HE, and *C6st-1* KO, mice treated or untreated with imiquimod on day 3 after initial treatment, stained using H&E. Yellow arrows indicate epidermis. **c** Graph shows epidermal thickness. Values are mean ± S.D. from four specimens. Statistical significance was determined using one-way ANOVA with Tukey's HSD test. **d** Dorsal skin sections from *C6st-1* WT, *C6st-1* HE, and *C6st-1* KO mice, treated or untreated with imiquimod for 1 day, were labeled with anti-K14 and anti-Ki67 antibodies. Nuclei were counter-stained with Hoechst33342. Arrows indicate Ki67-positive cells within K14-positive basal layer. **e** Ki67-positive cells in the K14-positive layer were quantified. Bars represent means ± S.D. from three specimens. Statistical significance was determined using one-way ANOVA with Tukey's HSD test. **f** Dorsal skin sections from *C6st-1* WT, *C6st-1* HE, and *C6st-1* KO mice treated or untreated with imiquimod on 3 day after initial treatment, were stained with anti-K14 antibody. Nuclei were counter-stained with Hoechst33342. Yellow arrows show K14-positive basal layer. **g** Thickness of the K14-positive basal layer was measured. Bars represent means ± S.D. from four specimens. Statistical significance was determined using one-way ANOVA with Tukey's HSD test. **h** Epidermis from *C6st-1* WT, *C6st-1* HE, and *C6st-1* KO mice, treated or untreated with imiquimod for 8 h, analyzed via immunoblotting using anti-phospho-EGFR and anti-tubulin. Graph under blots shows relative expression of phospho-EGFR in *C6st-1* WT, *C6st-1* HE, and *C6st-1* KO epidermis. Values represent the mean ± S.D. with n = 3 or 4 per group. Statistical significance was determined using one-way ANOVA with Tukey's HSD test. **i** Ki67-positive cells in epidermis at 8, 24, and 96 h after application of imiquimod, quantified as percentage of K14-positive cells. **j** Epidermal thickness was measured at 8, 24, and 96 h after imiquimod application. Values are mean ± S.D. with n = 3 per group. Statistical significance was determined using one-way ANOVA with Tukey's HSD test.

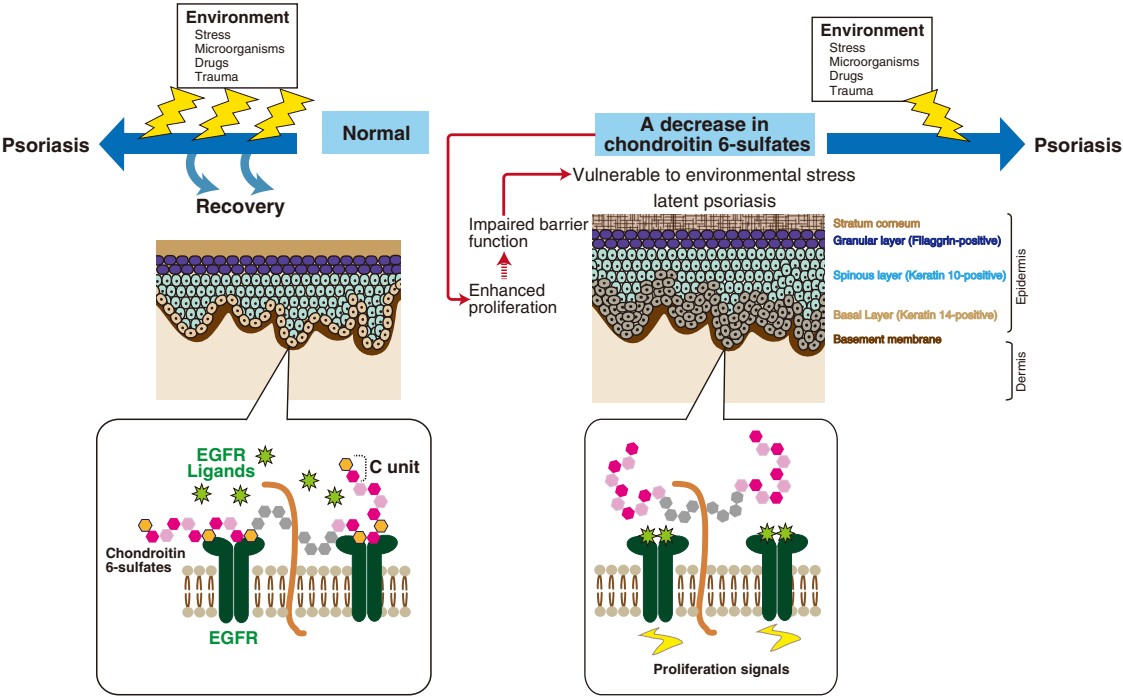

**Fig. 9 Chondroitin 6-sulfates and psoriasis.** Schematic representation of the skin. Mammalian skin consists of the epidermis and dermis, separated by a basement membrane. The epidermis is a stratified squamous epithelium that is composed of several cell layers. Resting on the basement membrane is the basal layer, consisting of K14-expressing proliferating cells. The basal layer gives rise to differentiated cell layers of the K10-expressing spinous layer, filaggrin-expressing granular layer, and the stratum corneum. Under normal conditions, keratinocytes in the K14-positive basal layer proliferate EGFR-dependently. Chondroitin 6-sulfate prevents keratinocytes from over-proliferation by interacting with EGFR. In contrast, a decrease in chondroitin 6-sulfate enhances keratinocyte proliferation signaling via activation of EGFR and impairs epidermal barrier function. When environmental factors trigger psoriasis, normal keratinocytes can maintain homeostasis; therefore, psoriasis does not develop from a single stimulus. Downregulation of chondroitin 6-sulfates causes para-inflammation in keratinocytes and may render keratinocytes vulnerable to environmental factors. Thus, a decrease in chondroitin 6-sulfates caused by mutations in the *C6st-1* or *Fam20b* genes may predispose individuals to psoriasis.

keratinocytes, our study shows that CS-C suppressed EGFR signaling, thereby regulating keratinocyte cell fate to maintain epidermal homeostasis.

Sulfation can occur at various positions within the chondroitin sulfate backbone. Because different sulfation patterns are involved in different biological processes, sulfation patterns are referred to as the sulfation code[28]. We previously reported that *C6ST-1* transgenic mice, which have upregulated CS-C, show impaired accumulation of aggrecan in perineuronal nets and lattice-like extracellular matrix structures composed of CS-PGs[29, 30]. Abnormal formation of perineuronal nets decreases the

accumulation of orthodenticle homeobox 2 (Otx2) and causes persistent cortical plasticity[29, 30]. Additionally, CS-C chains bound to aggrecan contribute to perineuronal nets and glial abnormalities in schizophrenia and bipolar disorders[31]. In this study, CS-C abrogated the EGFR-ERK1/2 and EGFR-STAT3 signaling pathway, suppressed keratinocyte proliferation, and directed epidermal differentiation. As shown in Fig. 5, chondroitin 4-sulfate (CS-A) did not inhibit activation of EGFR and proliferation of keratinocytes. Thus, epidermal homeostasis may require information from the chondroitin 6-sulfation code. Although our in situ proximity ligation assay and SPR assay

indicated that CS-C can interact with EGFR but not EGF (Fig. 6), we cannot exclude the possibility that other EGF ligands such as amphiregulin bind to CS-C. Further studies are needed to identify the proteins involved in the sulfation code of chondroitin sulfates in epidermal homeostasis.

This study will lead to improved understanding of the mechanisms involved in chondroitin 6-sulfate-mediated regulation of keratinocyte proliferation. Further studies will generate new therapeutic strategies, such as topical creams containing chondroitin 6-sulfate, for alleviating psoriasis.

## Methods

**Cell culture**. The human keratinocyte cell lines HaCaT (CLS #300493) and PSVK1 (JCRB1093) were obtained from the CLS cell line services at GmbH and JCRB cell bank, respectively. Normal human dermal fibroblasts were purchased from PromoCell GmbH (#C-12302) and Thermo Fisher Scientific (#A12592); psoriatic skin fibroblasts were obtained from DV Biologics (#AI001-F-PS-36p3). Fibroblasts were maintained in Dulbecco's modified Eagle's medium (DMEM) supplemented with 10% fetal bovine serum (FBS) and 1% antibiotic cocktail (10,000 μg/ml streptomycin and 10,000 units/ml penicillin). HaCaT cells were cultured in $Ca^{2+}$-free DMEM supplemented with 10% low-calcium FBS and DMEM containing 30 μM $Ca^{2+}$ [32]. PSVK1 cells were cultured in keratinocyte basal medium 2 containing 30 μM $Ca^{2+}$ and supplementMix (50 μM $CaCl_2$, 4 μl/ml bovine pituitary extract, 0.125 ng/ml recombinant human EGF, 5 μg/ml recombinant human insulin, 0.33 μg/ml hydrocortisone, 0.39 μg/ml epinephrine, and 10 μg/ml human transferrin (holo)) (PromoCell GmbH, Heidelberg, Germany). All cell lines were maintained at 37 °C and 5% $CO_2$.

**Animals**. *C6ST-1* knockout mice [33] were housed under specific pathogen–free conditions in an environmentally controlled, bio-clean room at the Institute of Laboratory Animals, Kobe Pharmaceutical University. The mice were maintained on standard mouse/rat chow and on 12-h light/12-h dark cycle. All experiments were conducted according to the institutional ethics guidelines for animal experiments and safety guidelines for gene manipulation experiments of Kobe Pharmaceutical University. All animal procedures were approved by the Kobe Pharmaceutical University Committee on Animal Research and Ethics.

**Preparation of mouse keratinocytes**. Post-natal day 0–5 neonates from *C6ST-1* heterogeneous (HE) mice were humanely euthanized using a $CO_2$ chamber, after which whole skin was peeled off. The peeled skin was washed with PBS, placed into a tube filled with ice-cold dispase digestion buffer [0.5 mg/ml dispase II (#383-02281, Wako)] in Keratinocyte Media 2 without growth factors (#C-20111, PromoCell), and incubated overnight at 4 °C. The epidermis was separated from the dermis using forceps, placed into a Petri dish containing keratinocyte Growth Media 2 supplemented with supplementMix (#C-20111, PromoCell), and the cell suspension was filtered. Cells were seeded at the density $0.5 \times 10^6$ cells/well and cultured at 37 °C and 5% $CO_2$.

**Real-time PCR**. Total RNA was extracted from the peeled frozen mouse skin as follows. Frozen mouse skin (20–50 mg) was crushed using a Precellys 24 tissue homogenizer with small beads (CK14) at 5000 rpm for 15 s, using 3 cycles with intervals of 20 s. Crushed tissues were homogenized in 1 ml TRIzole reagent (Invitrogen), and total RNA was isolated according to the manufacturer's protocol. Aliquots (1 μg) of total RNA were digested with 2 IU of RQ1 RNase-free DNase (Promega) for 30 min at 37 °C and then incubated for 10 min at 65 °C with stop solution (Promega). For reverse transcription, total RNA (0.75 μg) was treated with M-MLV reverse transcriptase (Invitrogen) using random primers (non-adeoxyribonucleotide mixture; pd(N)₉) (Takara bio Inc., Shiga, Japan). Quantitative real-time PCR was conducted using FastStart Essential DNA Green Master and a LightCycler 96 (Roche Applied Science) according to the manufacturer's protocols. Housekeeping gene glyceraldehyde-3-phosphate dehydrogenase (*GAPDH*) was used as an internal control for quantification. The primers used for real-time PCR are listed in Supplementary Table 3.

**Measurement of transepidermal water loss (TEWL)**. TEWL from the dorsal skin of 8-week-old mice was determined with a VAPO METER SWL5001 (Delfin Technologies Ltd.). Mean values of five measurements from each animal were determined. Data are expressed in g/hm² as means ± SD from WT and HE mice (n = 4), and KO mice (n = 3).

**Outside-in barrier function assay**. For Toluidine Blue staining, newborn mice were killed, washed in methanol for 5 min, washed in PBS, incubated for 15 min in 0.1% Toluidine Blue [34].

**Immunoblotting**. Skin and keratinocytes were solubilized using lysis buffer [1% NP40, 20 mM Tris-HCl (pH 7.5), 0.15 M NaCl, 1 mM EDTA, plus phosphatase

cocktails and protease inhibitor cocktails (Nacalai Tesque)]. Each sample was resolved using 10% SDS-polyacrylamide gels, transferred to PVDF membranes, and incubated overnight with anti-phospho-EGFR (Tyr1068) (D7A5) XP® rabbit monoclonal antibody (dilution ratio 1:1,000) (#3777, Cell Signaling), anti-EGFR (D38B1) XP® rabbit monoclonal antibody (dilution ratio 1:1,000) (#4267, Cell Signaling), anti-ERK1/2 (137F5) rabbit monoclonal antibody (dilution ratio 1:1,000) (#4695 S, Cell Signaling), anti-phospho-ERK1/2 (T202/Y204) (20G11) rabbit monoclonal antibody (dilution ratio 1:1,000) (#4376 S, Cell Signaling), anti-STAT3 rabbit polyclonal antibody (dilution ratio 1:1,000) (#9132 S, Cell Signaling), anti-phospho-STAT3 (Y705) rabbit polyclonal antibody (dilution ratio 1:1,000) (#9131 S, Cell Signaling), and anti-filaggrin antibody (dilution ratio 1:1,000) (#GTX37695, GeneTex). The bound antibodies were detected with anti-rabbit IgG conjugated to horseradish peroxidase (GE Healthcare). The full uncropped blot images are shown in Supplementary Fig. 1.

**Histology**. Tissues were fixed in 4% paraformaldehyde and embedded in paraffin. Tissues were sectioned at 5 μm, and stained with hematoxylin and eosin. Images were obtained with a Keyence BZ-8000 fluorescent microscope. For immunostaining, mouse dorsal and tail skin were cryopreserved in OCT compound, sectioned at 10 μm using a cryostat microtome (Leica CM3050S), hybridized to an anti-CS-C antibody (CS-56), and mounted on slides.

**Epidermal thickness measurement**. Images were acquired using the all-in-one BZ-X700 microscope (Keyence), and vertical distance between the skin surface and dermal epidermal junction (Fig. 1e(a), yellow lines) was measured using ImageJ (NIH). Measurements were performed on nine randomly selected tissue sections per animal. Three randomly chosen fields per tissue sections were used for the analysis and the mean of epidermal thickness was calculated for each section. Epidermal thickness of each animal was then calculated based on the mean values.

**Immunohistochemistry and immunocytochemistry**. Paraffin sections were stained using: antibody against Ki67 (mouse monoclonal; dilution ratio 1:250; #556003; BD Pharmingen), keratin 14 (1:250; rabbit polyclonal; Invitrogen), and keratin 10 (dilution 1:800; ab9026, clone DE-K10; Abcam). Cryosections were used for immunohistochemical analysis using anti-CS-C antibody (CS-56) (1:200; mouse monoclonal; Sigma-Aldrich) and anti-keratin 14 (1:500). For visualization, the sections were incubated with Alexa Fluor 488- or Alexa Fluor 594-conjugated secondary antibodies (Invitrogen), followed by embedding using mounting medium supplemented with Hoechst 33342. Images were acquired with a Zeiss LSM 700 confocal laser scanning system (Carl Zeiss Inc., Oberkochen, Germany) equipped with an inverted Axio observer Z1 microscope.

Cells were cultured on 13-mm cover glass (Matsunami Glass Ind., Ltd., Osaka, Japan) in a 24-well plate at the density of $1.0 \times 10^5$ cells/well. Cells were fixed with prechilled methanol for 20 min on ice and then washed with PBS containing 0.1% Tween 20. After blocking with PBS containing 1% BSA and 0.3% Triton X-100 for 1 h at 25 °C, cells were incubated overnight at 4 °C with the following antibodies: Ki67 (dilution 1:250), and anti-K14 monoclonal antibody (dilution 1:250). Cells were then washed with PBST and incubated with Alexa488-conjugated anti-rabbit IgG antibody (dilution 1:250; Molecular Probes, Eugene, OR) for detection of K14, or with Alexa594-conjugated anti-mouse IgG (dilution 1:250; Molecular Probes, Eugene, OR) for detection of Ki67. Hoechst33342 was used as nuclear counterstain. Images were acquired with a Zeiss LSM 700 confocal laser scanning system.

For the cell proliferation assay, HaCaT and PSVK1 cells were seeded into a 24-well plate (for immunofluorescence analysis) and 6-cm dishes (for immunoblotting). The cell proliferation assay proceeded as shown in Fig. 7a. Twenty-four hours post-seeding, cell starvation commenced via replacement of culture medium with serum-free growth medium (for HaCaT cells) or keratinocyte basal medium 2 containing 30 μM $Ca^{2+}$ but without supplement mix (for PSVK1 cells). After another 24 h incubation interval, the cell cycle was permitted to progress via addition of serum in the presence of 2.5 munits/mL of chondrotinase ABC and 5 μM PD153035. Cells were subjected to immunofluorescence analysis and immunoblotting at the indicated time (Fig. 7a).

**Proximity ligation assay (PLA)**. PLA was carries out using Duolink® In situ PLA® Probe. The interaction of chondroitin 6-sulfate with EGFR was detected using an anti-chondroitin 6-sulfete monoclonal antibody (dilution ratio 1:250 (primary keratinocytes and HaCaT), 1:2000 (PSVK1); clone CS-56, #C8035 Sigma-Aldrich) and anti-EGFR (D38B1) XP® rabbit monoclonal antibody (dilution ratio 1:1,000 (HaCaT), 1:500 (primary keratinocytes and PSVK1); #4267, Cell Signaling), respectively. Anti-rabbit IgG Minus and anti-mouse IgM PLUS were used as secondary antibodies.

**Disaccharide analysis of CS**. Isolation and purification of CS from mouse skin, epidermal splits, primary mouse keratinocytes, and human keratinocyte cell lines were performed [8, 9, 35]. Cells were homogenized in acetone, extracted with acetone three times, and air-dried thoroughly. The dried materials were digested with heat-activated actinase E (10% by weight of dried materials) in 0.1 M borate-sodium, pH 8.0, containing 10 mM $CaCl_2$ at 55 °C for 48 h. The samples were adjusted to 5% v/v in trichloroacetic acid and centrifuged. The resulting supernatants were extracted with diethyl ether three times to remove the trichloroacetic acid. The aqueous phase was evaporated to dryness, dissolved in 50 mM pyridine acetate, pH 5.0, and

subjected to gel filtration on a PD-10 column (GE Healthcare) using 50 mM pyridine acetate, pH 5.0, as an eluent. The flow-through fractions were collected and evaporated to dryness. Purified GAGs were digested with 5 munits of chondroitinase ABC at 37 °C for 4 h. Reactions were terminated by boiling for 1 min. The digests were derivatized with a fluorophore 2-aminobenzamide and then analyzed by high performance liquid chromatography.

**Surface plasmon resonance (SPR) analysis**. Real-time binding interaction studies were performed using a Biacore X100 (GE Healthcare, USA)[36]. Recombinant human EGFR protein (Fc chimera) (ab155726, Abcam) was immobilized on a CM5 sensor chip (GE Healthcare) using an amine-coupling method. Injection of EGFR was terminated when surface plasmon resonance reached ~ 3000 RU. For EGFR kinetics assays, EGF (Cat. No. 53003-018, Invitrogen) at 0, 0.813, 1.63, 3.25, and 6.5 μg/ml, or chondroitin sulfate C (CS-C) at 0, 15.6, 31.3, 62.5, and 125 μg/ml were sequentially injected at a flow rate of 30 μL/min for 120 s at 25 °C; dissociation time was set for 130 s (for EGF) or 180 s (for CS-C). Recombinant EGF protein (Fc chimera) (Cat. No. 10605-H01H, Sino Biological) was immobilized on a CM5 sensor chip. Injection of EGF was terminated when the surface plasmon resonance reached ~ 1600 RU. For EGF kinetics assays, EGFR protein (Fc chimera) at 0, 6.24, 12.5, 25, and 50 μg/ml, or CS-C at 0, 15.6, 31.3, 62.5, and 125 μg/ml were sequentially injected at a flow rate of 30 μL/min for 120 s at 25 °C; dissociation time was set to 3 min. Binding reactions proceeded in HBS-EP buffer (BIAcore) for EGF and EGFR, or Tris-HCl (pH 7.5) for CS-C. To evaluate the binding affinity, the equilibrium dissociation constant ($K_D$) was calculated for individual analytes using Biacore X100 Evaluation software (GE Healthcare), assuming a 1:1 binding ratio.

**Imiquimod Treatment**. Mice aged 6–10 weeks of age were anesthetized using isoflurane and denuded by shaving. Mice received a daily topical dose of 62.5 mg of 5% imiquimod cream (Beselna Cream; Mochida Pharmaceutical, Tokyo, Japan) on shaved lower backs for 1 day or 3 consecutive days. Consequently, the mice were humanely euthanized and tissues were harvested. Untreated upper back skin was used as a control. Skin adjacent to that collected for histology was used for isolation of total RNA and extraction of proteins.

**Statistics and reproducibility**. Data are expressed as mean ± standard deviation of the mean. Statistical significance was determined using Tukey–Kramer multiple comparison method and Student's $t$ test as noted in the figure legends. Statistical analyses were performed by KaleidaGraph version 4.5.1. For all analyses, $n =$ number of experiments, and differences are considered statistically significant at $p < 0.05$. All experiments were reproduced with similar results a minimum of three times, and the exact number of repetitions is provided in the figure legends.

**Reporting summary**. Further information on research design is available in the Nature Research Reporting Summary linked to this article.

## Data availability

The authors declare that the data supporting the findings of this study are available within the paper and its Supplementary information files or from the corresponding author on reasonable request. The source data for the figures presented in this study have been included in Supplementary Data 1.

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

## Acknowledgements

This work was supported in part by Grants-in-Aid for Scientific Research on Innovative Areas 23110003 (to H. K.) and Scientific Research (B) 25293014 and 20H03386 (to H. K.) and (C) 25460080 (to S. N.), and by the Support Program for Strategic Research Foundation at Private Universities 2012–2016 (to H. K.) from the Ministry of Education, Culture, Sports, Science, and Technology, Japan. We thank Arisa Inoue, Maho Kuroiwa, Shuichi Sato, and Kazuyuki Michikado for technical support.

## Author contributions

K.Kitazawa and S.N. performed the research and analyzed the data. K.Kadomatsu produced *C6st-1* knockout mice. S.N. and H.K. wrote the manuscript. H.K. conceived, designed, and coordinated the study. All authors reviewed the results and approved the final version of the manuscript.

## Competing interests

The authors declare no competing interests.
