## [Peer Review File · Communications Biology]

Reviewers' comments:

Reviewer #1 (Remarks to the Author):

This manuscript concerns the role of a specific isoform of chondroitin sulfate, namely chondroitin 6-sulfate and its influence on proliferation of keratinocytes. Using a variety of techniques, but focussing largely on heterozygote and knockout mice for the relevant 6-O-sulfotransferase, the authors report defects in differentiation, increased proliferation and increased susceptibility to imiquimod, a topical agent that triggers a psoriasis-like effect in mice. Further molecular studies infer that C6-S affects the epidermal growth factor receptor and its signaling, perhaps through direct interactions, though the precise mechanism remains unknown currently. This is an interesting and novel set of data, particularly since there may be a linkage to a common skin disease. The work is supported by a large set of figures and is well presented. While the presence of C6-S in the basement membrane underlying the epidermis has been known for around 30 years, its role(s) have remained largely unknown. This study provides a new and potentially important insight. There are a few major areas where further data or explanation seem necessary. In addition, there are some minor points, largely errors or omissions.

Major Points:

1. Inflammation. The authors point out the possible association of C6-S with the inflammatory skin disease, psoriasis. The title of the manuscript strongly links the two. In the discussion it is stated that the C6-S null mouse reports on the effects of the absence of this glycosaminoglycan in a "normal" context, but this is not shown. However, the authors do convincingly show that differentiation, including cell adhesion and the permeability barrier are compromised, resulting in increased transepidermal water loss. There is hyperproliferation and suprabasal mitosis. All of these features can be seen in inflammatory diseases. The authors should show whether or not the null mouse skin expresses inflammatory markers.
2. A striking feature of the data is how similar the heterozygote mice are to the full knockout of the C6-sulfotransferase. The authors' analysis shows that the amount of C6-S in heterozygote skin is approximately one-third of the wild type. Have the authors checked to assess whether CS-56 staining of the heterozygote dermal-epidermal junction is negative? This would be interesting in light of the accompanying data.
3. A single method, proximity ligation assay, is used to demonstrate an interaction between C6-S and the EGFR. As the authors state in the discussion, this does not rule out an interaction between C6-S and ligands, for example HB-EGF which binds heparin or heparan sulfate, so may bind C6-S also. A second independent analysis of the interaction(s) seems necessary, especially given the definitive statement provided in the abstract.
4. The authors use the term "thickness" in the context of K14 positive cells in the epidermis, but the meaning is unclear (e.g. lines 108, 199). Are there multiple K14+ve cell layers in the hyperproliferative state? Many of the figures are quite low power and some are weak, notably Ki67. Some improvement of these would be very beneficial. Moreover, the authors compare total epidermal thickness between the genotypes. The accuracy of this measurement depends on measuring samples in precisely the same way, e.g. from sections perpendicular to the skin surface. How was this assured?

Minor points:

1. The legend to Fig 4 is incomplete.
2. Line 100. The use of "expression" in the context of this sections seems inappropriate.
3. Line 117, typographical error "filaggrin"
4. In several places the authors use the phrase "expectedly" e.g. lines 139 and 142. This implies the authors are pre-judging the results and should be avoided.
5. Line 185 "more vulnerable" does not accurately describe the data included in this section. Accelerated response to the drug is shown here.
6. Line 509. There is reference to higher magnification images (that would be welcome) but they are not shown.

7. Line 578. Probably the authors mean "macroscopic" rather than "microscopic".

Reviewer #2 (Remarks to the Author):

This submission by Kitazawa and colleagues present data based on mice deficient for the enzyme chondroitin-6-O-sulfotransferase-1 (C6ST-1) to demonstrate that chondroitin-6-sulfate is involved in preventing pathologies like psoriasis. This link with psoriasis coming from increased susceptibility to the disease when FAM20B gene is poorly expressed and thus chondroitin 6-sulfate biosynthesis is impacted.

This paper brings a large amount of well-produced and analyzed data. It is likely bringing new and interesting information regarding the role of GAG in keratinocyte biology and pathology. Several questions though regarding some figures and discussed points are requiring corrections and answers to complete the provided information.

Major concerns:

1. The reason why there are two panels provided in Figure 1 D to illustrate WT skin, and why KET or KO skin is not illustrated is not explained.
2. Comments on the differences in thickness observed between Newborn and 8 weeks old mice skin should be added. Explanation of the procedure used to measure epidermal thickness is missing. Same for Figure 2 (b) providing information about the basal layer thickness. The Y-axis must be better renamed.
3. Whether the expression of the differentiation markers altered in HET and KO animals can be restored if CS-A or CS-C are provided in an in vitro system would help defining if such alterations are a direct consequence of the reduced chondroitin-6-sulfate availability or a consequence of the altered cell signalling.
4. Panels C and D in Figures 4 and 5 are exactly the same and bring confusion. The data provided on Figure 4 panel D about relative gene expression levels for inflammatory actors are expecting a confirmation at protein level, for instance by the use of ELISA assays.
5. In Figure 7, whether the increase in Ki67+ cells in HaCaT and PSVK1 cells due to treatment with Chase ABC is due to enhanced EGFR signalling should ideally be tested by mean of PD153035 compound.
6. The Discussion section is very short and weak. For instance, in the first paragraph, no link is discussed between inflammation and the expression of chondroitin sulfate. This is a crucial information that is lacking in this study.

Minor concerns:

1. Several typos must be corrected. e.g. line 54, page 2: Sulfotransferase instead of sulfotransferse.
2. line 102, page 4: K14 is not a marker for epidermal differentiation. It is a basal marker and rather linked to non-differentiated, proliferative keratinocytes.
3. Line 177: keratinocytes

Response to the Referee's Comments

The following revisions have been made. The cited page numbers etc are for the PDF file for your review, where revisions are indicated in red.

Reviewer #1 (Remarks to the Author):

This manuscript concerns the role of a specific isoform of chondroitin sulfate, namely chondroitin 6-sulfate and its influence on proliferation of keratinocytes. Using a variety of techniques, but focussing largely on heterozygote and knockout mice for the relevant 6-O-sulfotransferase, the authors report defects in differentiation, increased proliferation and increased susceptibility to imiquimod, a topical agent that triggers a psoriasis-like effect in mice. Further molecular studies infer that C6-S affects the epidermal growth factor receptor and its signaling, perhaps through direct interactions, though the precise mechanism remains unknown currently.

This is an interesting and novel set of data, particularly since there may be a linkage to a common skin disease. The work is supported by a large set of figures and is well presented. While the presence of C6-S in the basement membrane underlying the epidermis has been known for around 30 years, its role(s) have remained largely unknown. This study provides a new and potentially important insight. There are a few major areas where further data or explanation seem necessary. In addition, there are some minor points, largely errors or omissions.

Major Points:

1. Inflammation. The authors point out the possible association of C6-S with the inflammatory skin disease, psoriasis. The title of the manuscript strongly links the two. In the discussion it is stated that the C6-S null mouse reports on the effects of the absence of this glycosaminoglycan in a "normal" context, but this is not shown. However, the authors do convincingly show that differentiation, including cell adhesion and the permeability barrier are compromised, resulting in increased transepidermal water loss. There is hyperproliferation and suprabasal mitosis. All of these features can be seen in inflammatory diseases. The authors should show whether or not the null mouse skin expresses inflammatory markers.

Our response to major point #1

Thank you very much for your helpful comments. At the suggestion of the reviewer, we examined the expression level of pro-inflammatory cytokines, as shown in Fig. 4D. The expression levels of all cytokines examined in Fig. 4D were not affected by loss of C6-S in new-born dorsal skin. In adult dorsal skin, the gene expression levels of some pro-inflammatory cytokines were increased by loss of C6-S. These results suggest that ablation of *C6st-1* does not directly activate intrinsic pro-inflammatory signaling but might set the epidermis in a primed pro-inflammatory state because of impaired barrier function. (p. 6, l. 167-)

2. A striking feature of the data is how similar the heterozygote mice are to the full knockout of the C6-sulfotransferase. The authors' analysis shows that the amount of C6-S in heterozygote skin is approximately one-third of the wild type. Have the authors checked to assess whether CS-56 staining of the heterozygote dermal-epidermal junction is negative? This would be interesting in light of the accompanying data.

Our response to major point #2

At the suggestion of the reviewer, we examined the expression of C6-S in *C6st-1* HE and *C6st-1* KO mice in addition to *C6st-1* WT mice by using CS-56 antibody staining, as shown in Fig. 1D. C6-S was expressed beneath the keratin 14 (K14)-positive stem cells of the epidermal basal layer in *C6st-1* WT mice. In contrast, C6-S was not detected in epidermal basal layer of *C6st-1* HE and *C6st-1* KO mice. (p. 3, l. 93-)

3. A single method, proximity ligation assay, is used to demonstrate an interaction between C6-S and the EGFR. As the authors state in the discussion, this does not rule out an interaction between C6-S and ligands, for example HB-EGF which binds heparin or heparan sulfate, so may bind C6-S also. A second independent analysis of the interaction(s) seems necessary, especially given the definitive statement provided in the abstract.

Our response to major point #3

According to the reviewer's suggestion, we examined the interaction between C6-S and EGFR using SPR analysis. SPR analysis demonstrated that CS-C could bind to EGFR with a high affinity, while CS-A could not (Figure 6E). In addition, it was investigated whether CS-C could interact with EGF (Figure 6F). Neither CS-C nor CS-A bound to EGF. (p. 7, l. 202-)

4. The authors use the term "thickness" in the context of K14 positive cells in the epidermis, but the meaning is unclear (e.g. lines 108, 199). Are there multiple K14+ve cell layers in the hyperproliferative state? Many of the figures are quite low power and some are weak, notably Ki67. Some improvement of these would be very beneficial. Moreover, the authors compare total epidermal thickness between the genotypes. The accuracy of this measurement depends on measuring samples in precisely the same way, e.g. from sections perpendicular to the skin surface. How was this assured?

Our response to major point #4

According to the reviewer's suggestion, we have revised the sentence as follows: "the K14-positive cells in the basal layer were mitotically more active in newborn *C6st-1* HE and *C6st-1* KO epidermis than in *C6st-1* WT epidermis (Figure 2A(b))." (p. 4, l. 115-)

In addition, the data in Fig. 2B(a) were updated. Because few K14- and Ki67-positive cells were detected in the dorsal skin of adult mouse using immunofluorescence, we examined proliferation in the skin of adult mouse tail.

Moreover, the method for the measurement of skin thickness was described in the "Material and Method" section as follows (p. 11, l. 373-).

"Images were acquired using the all-in-one BZ-X700 microscope (Keyence), and vertical distance between the skin surface dermal epidermal junction (Figure 1E(a), yellow lines) was measured using ImageJ (NIH). Measurements were performed on nine randomly selected tissue sections per animal. Three randomly chosen fields per tissue sections were used for the analysis and the mean of epidermal thickness was calculated for each section. Epidermal thickness of each animal was then calculated based on the mean values."

Minor points:

1. The legend to Fig 4 is incomplete.

Our response to minor points #1

Corrected.

2. Line 100. The use of "expression" in the context of this sections seems inappropriate.

Our response to minor points #2

Deleted. (p. 4, l. 106)

3. Line 117, typographical error "filaggrin"

Our response to minor points #3

Corrected. (p. 4, l. 125)

4. In several places the authors use the phrase "expectedly" e.g. lines 139 and 142. This implies the authors are pre-judging the results and should be avoided.

Our response to minor points #4

At the suggestion of the reviewer, we have deleted the phrase "expectedly"
(p. 3, l. 80, p.5, l. 160, and p.5, l. 162)

5. Line 185 "more vulnerable" does not accurately describe the data included in this section. Accelerated response to the drug is shown here.

Our response to minor points #5

At the suggestion of the reviewer, we have revised the sentence as follows: "skin hypertrophy in the IMQ-treated *C6st-1* HE and *C6st-1* KO mice was accelerated compared with *C6st-1* WT mice." (p. 7, l. 241-,)

6. Line 509. There is reference to higher magnification images (that would be welcome) but they are not shown.

Our response to minor points #6

At the suggestion of the reviewer, magnified images were shown (panels a-2, b-2, c-2, d-2, e-2, f-2, g-2, h-2, and i-2) in Figure 3A. One of K10- and K14-double positive cells was shown in the inset.

7. Line 578. Probably the authors mean "macroscopic" rather than "microscopic".

Our response to minor points #7

Corrected.

Reviewer #2 (Remarks to the Author):

This submission by Kitazawa and colleagues present data based on mice deficient for the enzyme chondroitin-6-O-sulfotransferase-1 (C6ST-1) to demonstrate that chondroitin-6-sulfate is involved in preventing pathologies like psoriasis. This link with psoriasis coming from increased susceptibility to the disease when FAM20B gene is poorly expressed and thus chondroitin 6-sulfate biosynthesis is impacted.

This paper brings a large amount of well-produced and analyzed data. It is likely bringing new and interesting information regarding the role of GAG in keratinocyte biology and pathology. Several questions though regarding some figures and discussed points are requiring corrections and answers to complete the provided information.

Major concerns:

1. The reason why there are two panels provided in Figure 1 D to illustrate WT skin, and why KET or KO skin is not illustrated is not explained.

Our response to major points #1

At the suggestion of the reviewer, we examined the expression of C6-S in *C6st-1* HE and *C6st-1* KO mice in addition to *C6st-1* WT mice by using CS-56 antibody staining, as shown in Fig. 1D. C6-S was expressed beneath the keratin 14 (K14)-positive stem

cells of the epidermal basal layer in *C6st-1* WT mice. In contrast, C6-S was not detected in epidermal basal layer of *C6st-1* HE and *C6st-1* KO mice. (p. 3, l. 93-)

2. Comments on the differences in thickness observed between Newborn and 8 weeks old mice skin should be added. Explanation of the procedure used to measure epidermal thickness is missing. Same for Figure 2 (b) providing information about the basal layer thickness. The Y-axis must be better renamed.

Our response to major points #2

According to the reviewer's suggestion, we have added the comments as follows:

(p. 4, l. 97-)

“In addition, the epidermal thickness in newborn was thicker than that in 8-week-old mice. In this regard, it is reported that neonatal mice epidermal thickness in adult skin is thinner than that in infant skin because the epidermis reaches maximum thickness in the late embryo and cell proliferation gradually decreases within a few weeks postpartum¹⁵. Although there were differences in proliferation activity of keratinocytes between newborn and adult epidermis, *C6st-1* affected both newborn and adult epidermis.”

(p. 11, l. 373-)

The method for the measurement of skin thickness was described in the “Material and Method” section as follows.

“Images were acquired using the all-in-one BZ-X700 microscope (Keyence), and vertical distance between the skin surface dermal epidermal junction (Figure 1E(a), yellow lines) was measured using ImageJ (NIH). Measurements were performed on nine randomly selected tissue sections per animal. Three randomly chosen fields per tissue sections were used for the analysis and the mean of epidermal thickness was calculated for each section. Epidermal thickness of each animal was then calculated based on the mean values.”

Figure 2A(b)

Explanation was added in the legends as follows:

“Multiple random vertical lines perpendicular to the epidermal border were measured. From the mean thickness of these lines, epidermal thickness in K14-positive basal layer was calculated.”

The Y-axis was also renamed.

3. Whether the expression of the differentiation markers altered in HET and KO animals can be restored if CS-A or CS-C are provided in an *in vitro* system would help defining if such alterations are a direct consequence of the reduced chondroitin-6-sulfate availability or a consequence of the altered cell signalling.

Our response to major points #3

According to the reviewer’s suggestion, we have done the analysis and added the sentences as follows: (p. 4, l. 134-)

“We next examined the gene expression levels of differentiation markers (*K14*, *K10*, *Flg*, *Ivl*, and *Lor*) in primary keratinocytes (Figure 3G). The expression level of *K14* was significantly elevated in *C6st-1* KO keratinocytes compared with *C6st-1* WT keratinocytes. In addition, the expression level of *Ivl* in *C6st-1* KO keratinocytes was lower than that of *C6st-1* WT keratinocytes. However, altered differentiation marker expression levels could not be rescued by addition of CS-C to *C6st-1* KO keratinocytes. There were no significant differences in the expression levels of *K10*, *Flg*, and *Lor* between *C6st-1* WT and *C6st-1* KO keratinocytes (Figure 3G). Thus, gene expression levels of differentiation markers may be low during *in vitro* culture even in *C6st-1* WT keratinocytes. These results suggest that differentiation status of *C6st-1* KO keratinocytes is slightly but significantly altered by loss of CS-C. However, it is thought that CS-C does not directly regulate keratinocyte differentiation.”

4. Panels C and D in Figures 4 and 5 are exactly the same and bring confusion. The data provided on Figure 4 panel D about relative gene expression levels for inflammatory actors are expecting a confirmation at protein level, for instance by the use of ELISA assays.

Our response to major points #4

Figure 5C and D were deleted. The expression levels of some pro-inflammatory cytokines in the back skin of newborn and adult mice were measured as shown in Figure 4D. Expression levels of *Il1b*, *Il6*, *Il7*, and *Il23* were unaffected by *C6st-1* expression level in the dorsal skin of newborn mice (Figure 4D(a)). In adult mouse skin, expression levels of *Il6*, *Il7*, and *Il23* were elevated by loss of *C6st-1* (Figure 4D(b)). These results suggest that ablation of *C6st-1* does not directly activate intrinsic pro-inflammatory signaling but may impair epidermal barrier function, producing a state more primed for inflammation. (p. 6, l. 202-)
Thus, the protein expression level of each cytokine was not measured.

5. In Figure 7, whether the increase in Ki67+ cells in HaCaT and PSVK1 cells due to treatment with Chase ABC is due to enhanced EGFR signalling should ideally be tested by mean of PD153035 compound.

Our response to major points #5

We starved the cells prior proliferation assay to emphasize the effect of Chase ABC on the activation of EGFR. In this experimental system, Chase ABC increased the activation of EGFR and enhanced proliferation. These effects of Chase ABC were diminished by treatment with PD153035 as shown in figure 7B and C. (p. 7, l. 212-)

6. The Discussion section is very short and weak. For instance, in the first paragraph, no link is discussed between inflammation and the expression of chondroitin sulfate. This is a crucial information that is lacking in this study.

Our response to major points #5

According to the reviewer's suggestion, we have added the sentences as follows.
"In addition, hyperproliferation caused by loss of CS-C may indirectly disturb keratinocyte differentiation and thus epidermal barrier function (Figure 3). Such a sequence of events initiated by loss of CS-C may make keratinocytes more vulnerable to environmental stress (Figure 9). Thus, we consider that C6st-1KO keratinocytes exist in a primed activation state following inflammatory stimuli. A primed activation state may not necessarily involve increased expression of pro-inflammatory cytokines. This

primed state can be conceptualized as “readiness” state. Thus, C6st-1KO keratinocytes may rapidly respond or have a low-threshold response to inflammatory stimuli.

The mitotically active basal layer of the epidermis contains stem cells (basal cells). The basal layer produces and secretes extracellular matrix constituents, including CS-C, which contributes significantly to stem cell niche maintenance. As shown, CS-C localizes primarily to the basement membrane (Figure 1D). Contact of epidermal stem cells with basement membrane is of fundamental importance in regulating their proliferation and differentiation. Thus, the present study identifies CS-C as a novel regulatory stem cell niche component within skin.

One manifestation of aging is the decline in epidermal barrier function. In human skin, CS-C is specifically expressed in the basal lamina, and decreases an age-dependent manner²⁶. Results of the present study suggest that a decrease in CS-C impairs barrier function via dysregulating keratinocyte proliferation. Thus, age-related changes in CS-C may be causally associated with functional changes intrinsic to the aging process of human skin.” (p. 8, l. 262-)

Minor points:

1. Several typos must be corrected. e.g. line 54, page 2: Sulfotransferase instead of sulfotransferse.

Our response to minor points #1

Corrected. (p.2, l. 49)

2. line 102, page 4: K14 is not a marker for epidermal differentiation. It is a basal marker and rather linked to non-differentiated, proliferative keratinocytes.

Our response to minor points #2

Corrected. (p. 4, l. 110)

3. Line 177: keratinocytes

Our response to minor points #3

Corrected. (p. 7, l. 207)

Additional revisions

1. All other corrected or modified words, phrases, and sentences have been shown in red in the attached file for review.

REVIEWERS' COMMENTS:

Reviewer #1 (Remarks to the Author):

The authors have carefully revised this manuscript, which now contains new and helpful data. The study as a whole is an interesting and novel contribution, particularly as it may relate to inflammatory skin diseases.

The authors could note that there is a typographical error in Fig 9 (vulnerable).

Reviewer #2 (Remarks to the Author):

This revised manuscript has been considerably improved by the additions and corrections made in response to reviewers' comments.

Some typos are still present in Figures though:

Figure 2: Panel B a Hoechst is misspelled and should be replaced by Hoechst33340

Figure 9: Vulnerable, Keratine 14 are misspelled

The reason for (Epi) above Epidermis is unclear.

In the Discussion section, line 277: "decreases **in** an age-dependent manner"

Response to the Referee's Comments

The following revisions have been made. The cited page numbers etc are for the PDF file for your review, where revisions are indicated in red.

Reviewer #1 (Remarks to the Author):

The authors have carefully revised this manuscript, which now contains new and helpful data. The study as a whole is an interesting and novel contribution, particularly as it may relate to inflammatory skin diseases.

The authors could note that there is a typographical error in Fig 9 (vulnerable).

Our response

We have corrected the typographical error. Thank you very much.

Reviewer #2 (Remarks to the Author):

This revised manuscript has been considerably improved by the additions and corrections made in response to reviewers' comments.

Some typos are still present in Figures though:

Figure 2: Panel B a Hoechst is misspelled and should be replaced by Hoechst33340

Our response

We have corrected the typographical error. Thank you very much.

Figure 9: Vulnerable, Keratine 14 are misspelled

The reason for (Epi) above Epidermis is unclear.

Our response

We have corrected the typos and deleted "(Epi)" above Epidermis.

In the Discussion section, line 277: "decreases **in** an age-dependent manner"

Our response

We have added “in” in the sentence.

All other corrected or modified words, phrases, and sentences have been shown in red in the attached file for review.